



# Temperature Acclimation of Photosystem II Efficiency across Plant Functional Types and Climate

Patrick Neri[1], Lianhong Gu[2], Yang Song[1]

[1]University of Arizona

[2]Oak Ridge National Laboratory

*Correspondence to*: Y. Song ([chopinsong@arizona.edu](mailto:chopinsong@arizona.edu))

## Abstract

Modelling terrestrial gross primary productivity (GPP) is central to predicting the global carbon cycle. Much interest has been focused on the environmentally induced dynamics of photosystem energy partitioning and how improvements in the
description of such dynamics assist the prediction of light reactions of photosynthesis and therefore GPP. The maximum quantum yield of photosystem II ($\Phi_{PSIImax}$) is a key parameter of the light reactions that influence the electron transport rate needed for supporting the biochemical reactions of photosynthesis. $\Phi_{PSIImax}$ is generally treated as a constant in biochemical photosynthetic models even though a constant $\Phi_{PSIImax}$ is expected only for non-stressed plants. We synthesized reported $\Phi_{PSIImax}$ values from Pulse-amplitude modulated fluorometry measurements in response to variable temperatures across the
globe. We found that $\Phi_{PSIImax}$ is strongly affected by prevailing temperature regimes with declined values in both hot and cold conditions. To understand the spatiotemporal variability of $\Phi_{PSIImax}$, we analysed the dependence of the temperature acclimation of $\Phi_{PSIImax}$ on plant functional type (PFT) and habitat climatology. The analysis showed that temperature acclimation of $\Phi_{PSIImax}$ is shaped more by climate than by PFT for plants with broad latitudinal distributions or in regions with extreme temperature variability. There is a trade-off between the temperature range within which $\Phi_{PSIImax}$ remains maximal
and the overall rate of decline of $\Phi_{PSIImax}$ outside the temperature range such that species cannot be simultaneously tolerant and resilient to extreme temperatures. Our study points to a quantitative approach for improving electron transport and photosynthetic productivity modelling under changing climates at regional and global scales.

## Copyright statement



## 1 Introduction

Plant photosynthesis is central to the carbon cycle (Friedlingstein et al., 2020; 2022). Illuminating its complexity is needed to understand the carbon cycle-climate feedback and assess food production, biodiversity, and global ecosystem health.

Anthropogenic activities have induced a variety of rapid shifts in the earth's climate (IPCC, 2023) that impact photosynthesis and ecosystem services globally (Hatfield et al., 2018; Heinze et al., 2019). Factors such as temperature stress impact photosynthetic carbon assimilation differently across species and climates and have contributed to significant variability in terrestrial ecosystem productivity and carbon sequestration potential (Wahid et al., 2007; Ashraf & Harris, 2013; Heskel et al., 2016; Perez & Feeley, 2020; Kelly et al., 2021). Terrestrial biosphere models (TBMs) have examined and incorporated many

mechanisms of stress-induced photoinhibition of vegetation carbon assimilation (Berry & Bjorkman, 1980; Farquhar et al., 1980; Ball et al., 1987; Franks et al., 2017; Lawrence et al., 2019; Parazoo et al., 2020; Yin et al., 2021; Porcar-Castell et al., 2021). However, the inconsistency between physiological process-based modelled gross primary productivity (GPP) and inferred values via satellite and eddy-covariance flux towers continues to be an ongoing challenge (Dietze, 2014; Sun et al., 2019; Zhang & Ye, 2021).

Photosynthesis is typically separated into light-dependent reactions, which involve the absorption of light within the photosystem complexes (photophysical) and its conversion to oxidative-reductive energy (photochemical), and carbon reactions that further utilize the photochemical energy as preserved in energy currency ATP and reducing power NADPH to perform carbon fixation through the Calvin-Benson cycle (biochemical) (Whatley et al., 1963; Kamen, 1963; Stirbet et al., 2019; Buchanan, 2016). Process-based models of net photosynthetic $CO_2$ assimilation generally centre on the simulation of

the biochemical reactions that are coupled with gas exchange via stomata (Farquhar et al., 1980; Ball et al., 1987; Lin et al., 2012; Yin et al., 2021). These models implement temperature regulation on biochemical kinetics (Rogers et al., 2017) and environmental dependence of stomatal conductance (Buckley, 2017), allowing mechanistic descriptions of the impact of water, temperature, and $CO_2$ concentrations on the dynamics of biochemical reactions. However, light reactions, especially mechanistic regulation by environmental factors, are treated less extensively.

Photophysical reactions control the dissipation of absorbed energy among different pathways, including fluorescence, photochemistry (PQ), constitutive heat dissipation, and non-photochemical quenching (NPQ). These pathways are subject to the constraint of energy conservation. NPQ can be further separated into energy-dependent and energy-independent mechanisms. The energy-dependent NPQ, also known as reversible NPQ, quickly relaxes after removing illumination and is connected to the xanthophyll cycle (Johnson et al., 1993; Demmig-Adams & Adams, 2006). The energy-independent NPQ,

also known as sustained NPQ, relaxes at longer time scales and can operate seasonally or even inter-annually with the mitigation of environmental stresses (e.g., temperature, water), and involves protein accumulation and photoinhibition (Demmig-Adams & Adams, 2006; Takahashi & Badger, 2011; Tietz et al., 2017). The PQ pathway transports electrons and protons to produce NADPH and ATP, consequently regulating the carbon reaction rates of photosynthesis. This pathway is typically quantified by the fraction of available photosystem II (PSII) reaction centers (qL) for charge separation after receiving



excitation energy. When the NPQ pathway is completely disengaged (NPQ = 0) and all PSII reaction centers are open (qL=1) under non-stress conditions, plants operate with maximum light use efficiency (LUE) for biochemical carbon assimilation, with an idealized maximum quantum yield of photosystem II ($\Phi_{PSIImax}$) of 0.75-0.85 (Kitajima & Butler, 1975; Bjorkman & Demmig, 1987; Genty et al., 1989; Corcuera et al., 2011). This value is generally treated as an environmentally independent constant in photosynthesis models (e.g., 0.85 in the Community Land Model (Lawrence et al., 2019)).

However, $\Phi_{PSIImax}$ can be irreversibly downregulated due to plant energy-independent NPQ acclimation to temperature and other environmental stresses, especially extreme temperature, or as a result of photodamage to reaction centers (i.e., qL is less than 1 even when plants are fully dark-adapted (Porcar-Castell, 2011)). This downregulation can induce a significant reduction to vegetation productivity (Havaux et al., 1992; Oberhuber & Edwards, 1993; Lu & Zhang, 1999; Murata et al., 2007; Ferguson et al., 2020; Kunert et al., 2022) but has not been mechanistically parameterized in most photosynthesis

models. Moreover, this impact of stress on the light reactions has been found to be highly variable among plant species across diverse regions (Corcuera et al., 2011; Marias et al., 2016; Perez & Feeley, 2020). In the Amazon, extreme temperature-induced reduction in $\Phi_{PSIImax}$ is irreversible and currently decreasing the productivity of tropical forests, with large variability in response among forest species (Tiwari et al., 2021). In addition, distinct differences in temperature tolerance and resilience of $\Phi_{PSIImax}$ values are also found among the same species growing in different habitats (Corcuera et al., 2011; Fadrique et al.,

2022). To better assess the tolerance and resilience of plant photosynthesis to more extreme climate change, there is an urgent need for a more mechanistic understanding and parameterization of the environments' impact on photosystem efficiency and its variability across species and habitats (McCallum et al., 2013; Dusenge et al., 2019; Fadrique et al., 2022).

The most common method for determining the various quantum yields of energy dissipation pathways is via monitoring Chlorophyll *a* fluorescence (ChlaF). Pulse-Amplitude Modulated (PAM) fluorometry is a routine non-invasive

method for investigating energy partitioning among the four dissipation pathways (Kitajima & Butler, 1975; Bjorkman & Demmig, 1987; Klughammer & Schreiber, 2008; Porcar-Castell, 2011; Lazár, 2015), and can serve as a bridge to modelling mechanistic partitioning of adsorbed light energy at the leaf level (Gu et al., 2019; Han et al., 2022). A dark-adapted, homeostatic plant minimizes the energy partitioning to the thermal and non-photochemical dissipation pathways, leading to the maximum light allocation fraction to the photochemical pathway (Klughammer & Schreiber, 2008). PAM fluorometry

experiments identify $\Phi_{PSIImax}$ by quantifying the ratio of the increase of fluorescence yield during a saturation pulse (Fv) to the maximal fluorescence yield of a dark-adapted sample (Fm). At the canopy level, ground-based and satellite Solar-induced ChlaF (SIF) measurements (Mohammad et al., 2019) have been increasingly integrated or assimilated to facilitate regional and global-scale GPP prediction (Lee et al, 2015; Norton et al, 2018; Norton et al., 2019; Bacour et al., 2019a; Bacour et al, 2019b; Yang et al, 2021). The accuracy of this model-SIF data integration depends on the ability of these models to represent GPP-

SIF relationships at leaf and canopy levels. Sun et al. (2023a; 2023b) highlighted the complexity of fully describing the many leaf and canopy level factors at play in the SIF-GPP relationship. Parazoo et al. (2020) examined seven TBMs that included SIF-based photosynthetic parameterization and found that much of their discrepancy may be tied, among other things, to the





need for better descriptions of leaf mechanisms of energy partitioning under environmental stress; others have pointed out similar areas of needed research (Rogers et al., 2017; Kumarathunge et al., 2019).

Our previous effort (Gu et al., 2019) has modelled the leaf-level SIF-GPP dynamics as a function of NPQ, qL, $\Phi_{PSIImax}$, and absorbed photosynthetically active radiation (APAR). Our study pointed out a need for mechanistic descriptions of how NPQ, qL, and $\Phi_{PSIImax}$ respond to environmental conditions to truly predict environmental regulation on GPP-SIF relationships at the leaf level. Built upon this study, here we present a novel model of $\Phi_{PSIImax}$ acclimation to temperature variation by collecting and applying a global-scale database of published PAM measurements. This effort focuses on $\Phi_{PSIImax}$, as modeling

temperature regulation on $\Phi_{PSIImax}$ is important not only for assessing extreme temperature impacts on the maximum electron transport rate ($J_{max}$) in biochemical reactions-centered photosynthesis models but also for resolving coupled SIF and GPP relationship in SIF-incorporated photosynthetic models. Characterizing the temperature response of $\Phi_{PSIImax}$ can thus allow a connection to most of the mechanisms needed to capture the temperature feedback of photosynthetic light reactions on the SIF-GPP dynamic (Sun et al., 2023a).

In this study, we developed specific temperature acclimation functions of $\Phi_{PSIImax}$ for 12 plant functional types (PFTs) commonly used in TBMs and determined temperature 'tolerance' and 'resilience' parameters for $\Phi_{PSIImax}$. In addition, the climatological impact on the temperature tolerance and resilience of $\Phi_{PSIImax}$ are also examined via creating a climatology index and incorporation into the original parameterization. Finally, we identified specific geographical locations where climate significantly affects PFT-specific temperature-$\Phi_{PSIImax}$ relationships.

**2 Data & Methods**

**2.1 PAM fluorometry data collection**

We quantified the impact of temperature on $\Phi_{PSIImax}$ by collecting 380 published studies with Fv/Fm data measured using the PAM fluorometry method from four publication repositories (Fig. 1a). To isolate temperature dependence from other external regulators of $\Phi_{PSIImax}$, we mined and selected data from studies that provided cohesive descriptions of temperature for the

relevant measurements and excluded the effects of other confounding variables (e.g., water, nutrient, light stress). Following these guidelines, a total of 104 studies out of the 380 publications were finally selected. Once selected, the measurements of Fv/Fm were either directly recorded (for tabular and text reporting of Fv/Fm values) or extracted from graphics using a web-based extraction tool (Rohatgi, 2021). The corresponding experimental temperature, together with the study location, measurement techniques, duration of the temperature exposure, taxonomic description, and other factors of interest, were

recorded (Data set S1). As reporting of temperature was not consistent across studies, we used three methods to identify experimental temperature: (1) for publications that utilized a diurnal description of temperature, the diurnal mean temperature was used as a proxy measurement temperature, as determined by the average of the minimum and maximum reported values. (2) For studies performed in uncontrolled temperature environments, the mean temperature experienced by the plant during the experiment period was used. (3) If a study lacked a well-described reporting of specific experimental temperature, the





mean temperature during the experimental period was collected from the 0.5° x 0.5° global atmospheric forcing dataset, CRUNCEP v.7 (Viovy et al., 2018).

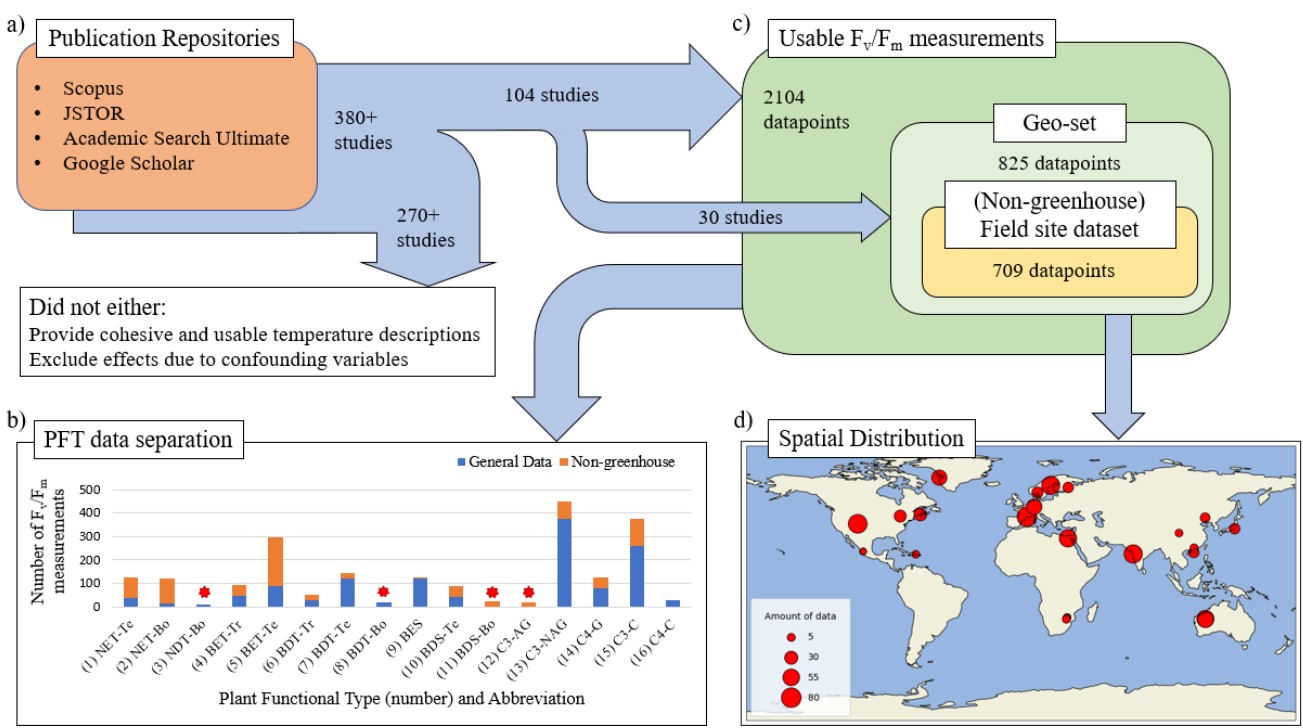

Figure 1. Data acquisition, selection, and composition pipeline. (a) A summary of the rule-specific procedure of mining and selecting PAM fluorometry data from publication repositories; (b) The number of available PAM data for the 16 chosen PFTs,
being (1) needleleaf evergreen temperate tree (NET-Te), (2) needleleaf evergreen boreal tree (NET-Bo), (3) needleleaf deciduous boreal tree (NDT-Bo), (4) broadleaf evergreen tropical tree (BET-Tr), (5) broadleaf evergreen temperate tree (BET-Te), (6) broadleaf deciduous tropical tree (BDT-Tr), (7) broadleaf deciduous temperate tree (BDT-Te), (8) broadleaf deciduous boreal tree (BDT-Bo), (9) broadleaf evergreen shrub (BES), (10) broadleaf deciduous temperate shrub (BDS-Te), (11) broadleaf deciduous boreal shrub (BDS-Bo), (12) C3 Arctic grass (C3-AG), (13) C3 non-Arctic grass (C3-NAG), (14) C4
grass (C4-G), (15) C3 crop (C3-C), and (16) C4 crop (C4-C). A PFT that has less than 30 data is marked by a red star.; (c) The definition of sub-datasets according to the availability of geographic information for the data and PAM experimental methodology; (d) Spatial distribution of the 'geo-set' of data and the number of measurements from each study represented by the circle size.

In total, 2104 measurements from 104 sites were recorded in the final database. All measurements included plant taxonomic descriptions, culminating in 146 distinct species, from 46 different family groups and 29 orders. We grouped these measurements into 16 PFT-specific sub-datasets (Fig. 1b). These sub-datasets were analysed for PFT-specific $\Phi_{PSIImax}$ responses to temperature change. In addition, within the 2104 total measurements, a subset of 825 measurements from 30 publications included explicit latitude and longitude information. This 'geo-set' of data covers diverse geographic regions
across a bounding range of [107°W, 35.4°S, 140°E, 69.25°N] (Fig. 1d). For data within this geo-set, 709 measurements were collected from in-the-wild plants located in natural environments (non-greenhouse) (Fig. 1c). This geo-specific and climate-





affected sub-dataset, herein called field site sub-dataset, was used to assess the effects of climatological temperature on the response behaviour of plant $\Phi_{PSIImax}$ to temperature change.

## 2.2 Parameterizing temperature regulation on $\Phi_{PSIImax}$ for each plant functional type

We employ the PFT-specific sub-datasets to parameterize a general temperature acclimation function of $\Phi_{PSIImax}$ for all data, and 12 PFT-specific temperature acclimation functions. We quantified the temperature tolerance and resilience of $\Phi_{PSIImax}$ for each PFT based on the corresponding parameterized temperature acclimation function.

### 2.2.1 Selecting the fitting function and quantifying temperature tolerance and resilience

Features of a function desirable for the parameterization of the gathered data are:

• Capturing the characteristics of the temperature response of $\Phi_{PSIImax}$

   • Continuous across the full range of temperatures at a global scale

   • Well suited for further refinement of parameters with additional data through fitting methods

   • Physically interpretable parameters

With these characteristics in mind, the selected parameterization scheme was a rectangular function of temperature with the

form:

$$\Phi_{PSIImax} = F_v/F_m = f(T) = a\frac{1}{2}\left(erf\left(\frac{T-m_1}{s_1}\right) + erf\left(\frac{m_2-T}{s_2}\right)\right) \qquad \text{Eq. 1}$$

Where T is the temperature in degrees Celsius and erf is a special class of the sigmoid function called the error function (erf):

$$erf(z) = \frac{2}{\sqrt{\pi}}\int_0^z e^{-t^2}dt \qquad \text{Eq. 2}$$

Each of the five parameters has a physical interpretation. The *a* parameter is the maximal value of $\Phi_{PSIImax}$ under no limitation

from temperature or other environmental factors. The $m_1$ ($m_2$) parameters mark the temperature at which $\Phi_{PSIImax}$ has declined by 50% of the *a* value (also called $T_{50}$, see Marias et al., 2016; Leon-Garcia & Lasso, 2019; Perez & Feeley, 2020) at cold (hot) temperatures, with the units of [°C]. The $s_1$ ($s_2$) parameters have units of [°C] and indicate the slope of peak $\Phi_{PSIImax}$ decrease at the $m_1$ ($m_2$) temperature values, which can be thought of as a plant's resilience to cold (hot) temperatures, as a smaller $s_1$ ($s_2$) means a more rapid decline in $\Phi_{PSIImax}$ under cold (hot) temperatures, indicating a lower resilience to cold (hot)

temperatures.

Furthermore, the temperature range within which the predicted $\Phi_{PSIImax}$ remains steady at its maximum (*a*) can be estimated by creating a linear combination of $s_i$ & $m_i$ parameters (Eqs 3-4) shown below. The lower ($T_{MC}$) and upper ($T_{MH}$) bounds of this temperature range are referred to as a plant's tolerance to cold and hot temperatures, respectively.

$$T_{MC} = m_1 + 2s_1 \qquad \text{Eq. 3}$$

$$T_{MH} = m_2 - 2s_2 \qquad \text{Eq. 4}$$



$\Phi_{PSIImax}$ starts to decrease from its peak value as the temperature drops below $T_{MC}$ or heats above $T_{MH}$. We determined the best fit for the parameters in Eq. 1 using a variation of the Levenberg-Marquardt method (LMFIT package version 1.0.3) (Newville et al., 2014). The model performance was indicated using the coefficient of determination ($R^2$).

### 2.2.2 Model fitting and parameter constraints

To ensure that the model could consistently capture the pattern shown in the gathered data, a cross-validation test for parameterizing Eq. 1 was performed. The full dataset of measurements using all PFTs underwent a permutation of order, and 70% of the data was selected for a calibration test. The resulting parameterization was then used to predict the $\Phi_{PSIImax}$ value based on temperature for the remaining 30% of the dataset. This process was iterated 1000 times, with the $R^2$ recorded. This test was performed twice to check whether there were enough iterations to properly represent the general statistical tendency.

Available data within each PFT-specific sub-dataset may cover different temperature ranges with associated variability in $\Phi_{PSIImax}$. This data characteristic may hinder the reliable estimation of the five parameters with the fitting algorithm for the following reasons. First, $\Phi_{PSIImax}$ outliers at low (high) temperatures can have outsized impacts on estimating $m_1$ ($m_2$) and $s_1$ ($s_2$) parameters, leading to prediction biases for $\Phi_{PSIImax}$ at low (high) temperatures. Furthermore, if the data used in model fitting did not cover a wide range of temperatures, or the minimum $\Phi_{PSIImax}$ value was not less than $a/2$,

parameters cannot be estimated accurately. To avoid these prediction biases and make comparable the fitted values of parameters for each PFT-specific temperature response function of $\Phi_{PSIImax}$, we imposed unified constraints on each parameter's range (Table A1) using a Monte Carlo scheme (See Appendix A).

Finally, we applied paired $\Phi_{PSIImax}$ and temperature data for each PFT to fit the PFT-specific temperature-$\Phi_{PSIImax}$ function (Eq. 1). To mitigate overfitting and ensure the available temperature-$\Phi_{PSIImax}$ measurement pairs covered enough

temperature variability for quantifying the decline outside of a central range of temperatures, we desired at least 30 data pairs and also a decline of $\Phi_{PSIImax}$ values by at least 10% from the maximum values. Only 12 of the 16 PFTs (Fig. 1b) were fully considered in the resulting analysis (Sect. 3.1), with some resultant cold (hot) parameters being treated with caution.

### 2.3 Parameterizing climatology influence on the temperature-$\Phi_{PSIImax}$ relationship

To test the hypothesis that climatological temperature regulates the temperature tolerance and resilience of $\Phi_{PSIImax}$, we

identified climatology temperature metrics for the field site sub-dataset (Sect. 2.3.1) and quantified their capacity to reduce the prediction deficiency of the temperature-$\Phi_{PSIImax}$ model (Sect. 2.3.2). Based on the results, we incorporated the metrics via a linear combination (Sect. 2.3.3) into a Climatology Temperature Index (CTI). This index was then incorporated into the parameterization of Eq.1 (Sect. 2.3.4). The fitting results of the revised model were compared to the corresponding PFT-specific model results. Finally, we identified where prediction deficiency was improved by the CTI-informed parameterization

and the climatology's effect on the temperature-$\Phi_{PSIImax}$ relationship was important to consider (Sect. 2.3.5).





### 2.3.1 Assessment of climatology temperature metrics

There were 25 locations ranging from [107°W, 35.4°S, 140°E, 69.25°N] within the field site sub-dataset (Fig. 1d). For each location, we used hourly temperature from 0.5° x 0.5° CRUNCEP v.7 forcing dataset (Viovy et al., 2018) between 1985 and 2016 to quantify three climatology temperature metrics: the average annual temperature (AAT), the summer median experienced temperature (SMET), and the winter median experienced temperature (WMET), respectively. To isolate a summer (winter) season based on temperature, a rolling 3-month average was performed to find the warmest (coldest) consecutive three months. From these three months, the median temperature served as the location's summer (winter) median experienced temperature (SMET, WMET) over 30 years (Fig. 2a). In addition, an AAT for each location was estimated by first calculating the annual mean temperature of each year and then averaged over all 30 years (Fig. 2a).

### 2.3.2 Quantifying climatology impact on prediction efficiency

To examine how climatology metrics affect the prediction efficiency of the developed PFT-specific model, an Aligned Rank Transform Analysis of Variance (ART ANOVA) was performed. We calculated the residues ($X$) between collected $\Phi_{PSIImax}$ values ($\Phi_{PSIImax,O}$) and predicted $\Phi_{PSIImax}$ values ($\Phi_{PSIImax,P}$) given by a specific temperature-$\Phi_{PSIImax}$ function parameterization (Eq. 5).

$$X = \Phi_{PSIImax,P} - \Phi_{PSIImax,O}$$ Eq. 5

We then analysed $X$ with ART ANOVA. ART ANOVA is a nonparametric test of data variation with multiple factors. It allows a determination of the contribution of variance by each factor and the interaction effect of multi-factors when assumptions of equal sample sizes within each factor level needed for conventional ANOVA parametric tests are not met (Leys & Schumann, 2010). Here the divisions of each climatology temperature metric level were determined via the Freedman-Diaconis rule (Freedman & Diaconis, 1981) as follows.

$$\text{Bin Width} = 2\,\text{IQR} \times n^{-1/3}$$ Eq. 6

Where $n$ is the number of residues used in the test and IQR is their interquartile range.

Following Wobbrock et al. (2011)'s ART ANOVA analysis procedure, we first performed a preprocessing step to align the $X$ of each $\Phi_{PSIImax}$ value in the field site sub-dataset for each main effect of the predictors (SMET, WMET, AAT), as well as their two-ways and three-ways effects, using Eqs. B1-7 (see Appendix B). Second, the aligned $X$ values for each effect were then ranked from smallest to largest, with ties between $M$ numbers of $X$ values resulting in the sum of the ranks divided by $M$. Finally, a standard ANOVA test was performed on the ranks of the aligned $X$ values for each effect and their combinations, respectively. Here all main predictor effects and their two-way and three-way interaction effects were induced at each instance of ANOVA analysis, while only the total sum of the squares of errors of the tested effect was kept. Also, the total sum of squares of the residuals from each ANOVA analysis was recorded and averaged to represent the final sum residual that failed to be explained by the three climatology temperature metrics and their interaction effects. The resulting total sum of the squares is each effect's sum of squares and the residual term. The ratio of each effect's sum of squares to the total sum





of squares is a measure of the explained prediction error resulting from a specific temperature-$\Phi_{PSIImax}$ model by SMET, WMET, AAT, and their interactions, respectively.

Besides performing the above ART ANOVA analysis to explain the contribution of climatological temperature to the prediction error of the developed 12 PFT-specific temperature-$\Phi_{PSIImax}$ functions, we also performed the second ART ANOVA analysis to examine the contribution of three temperature metrics and their interactions to the prediction residues by the general temperature-$\Phi_{PSIImax}$ function derived using all data in the field site dataset. The results of two ART ANOVA were then compared.


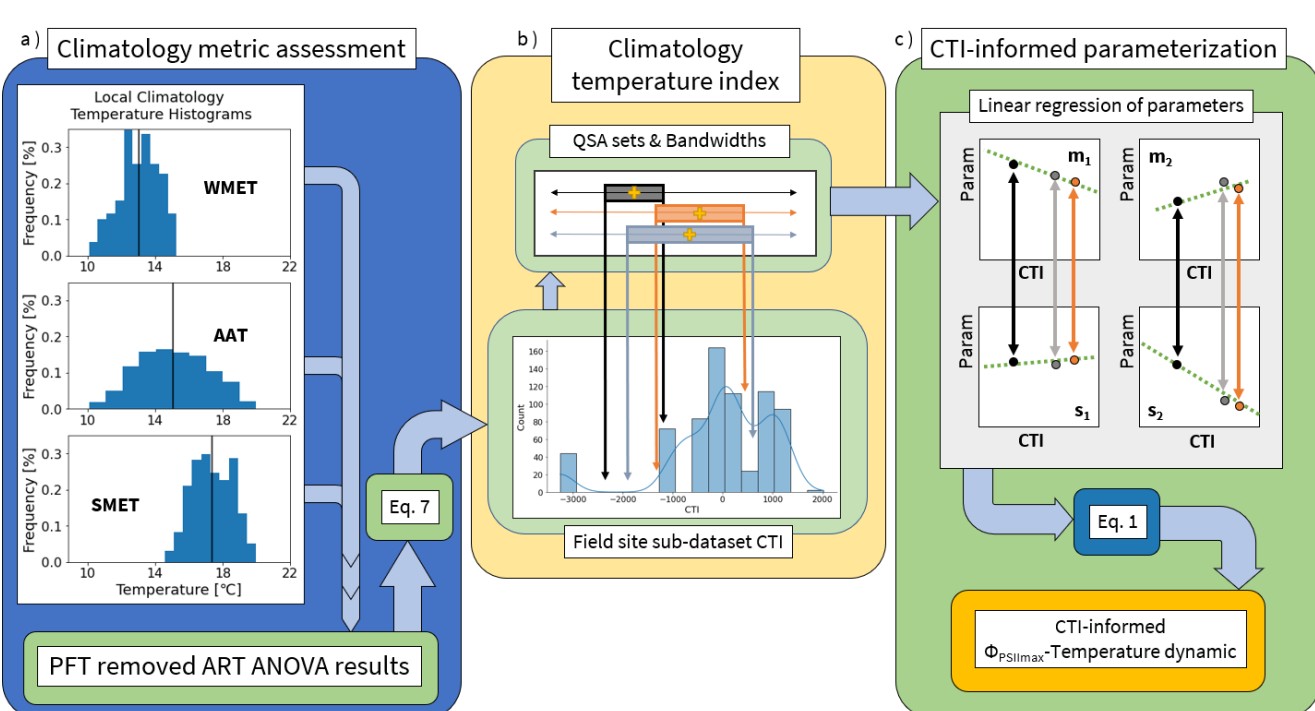

Figure 2. A pipeline of incorporating climatological temperature's effect on the modelled parameterization of temperature tolerance and resilience of vegetative $\Phi_{PSIImax}$, including (a) quantifying the histogram frequency distribution of three climatological temperature metrics (WMET, SMET, AAT) within the field site sub-dataset and their contributions to the
prediction deficiency estimated by the developed PFT-specific temperature-$\Phi_{PSIImax}$ functions using ART ANOVA. WMET and SMET refer to the median experienced temperature in the winter and summer, respectively, while AAT refers to the annual average temperature calculated using 0.5° x 0.5° CRUNCEP v.7 hourly temperature dataset over 1985-2016. (b) calculating the Climatology Temperature Index (CTI) for all data in the field site sub-dataset using results of ART ANOVA and Eq. 7, with the resulting distribution of CTI values shown in a histogram, and the Quantile System Approach (QSA)-based data
grouping and parameters ($m_1, m_2, s_1, s_2$) estimation. Here, QSA refers to using multiple bandwidths (shown as different color boxes) to group data for fitting Eq. 1, varying the range of the selected data across the whole dataset, and recording the central CTI (shown as gold cross) and resulting parameters. (c) The CTI-informed parameterization of temperature-$\Phi_{PSIImax}$ dynamic, which linearly regressed each of the four parameters ($m_1, m_2, s_1, s_2$) on the corresponding central CTI values resulting in a CTI-informed parameter estimation for the temperature-$\Phi_{PSIImax}$ function.






### 2.3.3 Climatology Temperature Index

To create a single predictor index that incorporates the effect of three temperature metrics on predicting the temperature response function of $\Phi_{\text{PSIImax}}$, the climatology temperature index (CTI) was created (Fig. 2b). This index, which was associated with each $\Phi_{\text{PSIImax}}$ measurement in the field site sub-dataset, was determined by creating a linear combination of SMET,

WMET, and AAT (Eq. 7) based on the results from the above ART ANOVA analysis. For each $\Phi_{\text{PSIImax}}$ measurement $p$ in the field site sub-dataset, its CTI value is given as

$$\text{CTI}_p = a_1\text{SMET}_p^* + a_2\text{WMET}_p^* + a_3\text{AAT}_p^* + a_4\big(\text{SMET}_p^* \times \text{WMET}_p^*\big) + a_5\big(\text{SMET}_p^* \times \text{AAT}_p^*\big) +$$
$$a_6\big(\text{WMET}_p^* \times \text{AAT}_p^*\big) + a_7\big(\text{SMET}_p^* \times \text{WMET}_p^* \times \text{AAT}_p^*\big) \qquad \text{Eq. 7}$$

Where $a_L$ ($L$=1, 2, …7) is the relative contribution of each climatological temperature metric and their interactions to variations

in prediction residues by each temperature-$\Phi_{\text{PSIImax}}$ function (PFT-specific or general) as shown in Fig. 5. The * denotes the deviation from the mean of all the respective climatology index values within the field site dataset.

### 2.3.4 Quantile system approach for parameterizing climatology's effect on temperature resilience and tolerance of $\Phi_{\text{PSIImax}}$

To incorporate the climatology factors into the parameterization of the temperature-$\Phi_{\text{PSIImax}}$ function (Eq. 1), we quantified the

dependence of the parameters ($m_1$, $m_2$, $s_1$, $s_2$) on CTI using the field site sub-dataset. Ideally, the field site sub-dataset would cover diverse climatological temperature conditions, be distributed consistently across the full global range of CTI values, and contain data from all PFTs, but this is not the case. The available 709 measurements represent a limited, non-uniform range of climatology temperature metrics (Histogram distribution of data in Fig. 2b). We overcome this data limitation using the quantile system approach (QSA), which was developed to navigate the small sample size and inconsistent CTI value

distribution by performing the following three steps.

First, to identify ranges of CTI upon aggregation can overcome the non-uniform distribution of CTI values, we ranked from least to greatest all field site sub-dataset $\Phi_{\text{PSIImax}}$ and their corresponding experimentally measured temperature values based on the CTI values at the location of the associated studies. A bandwidth (B) is defined as a quantile range of CTI values in the field site sub-dataset and recorded as a percentile. The corresponding $\Phi_{\text{PSIImax}}$ and experimentally measured temperature

within the bandwidth B were selected and composed a CTI-labelled sub-dataset. The central value of the range of CTI selected in this manner, here being the average of the upper and lower bounds within B, was used as a description of CTI for the corresponding CTI-labelled sub-dataset. We then used the sub-dataset and its central CTI value to define a 'QSA set', which then is connected to the parameters of Eq. 1 through fitting. This bridges the parameters ($m_1$, $m_2$, $s_1$, $s_2$) to the QSA set's associated climatology. The maximum value of $\Phi_{\text{PSIImax}}$ parameter (a) in Eq. 1 was assumed to be independent of CTI and kept

the same as the fitted value based on the PFT-specific parameterizations in Sect. 2.2. Four bandwidths (B = 66%, 50%, 33%,





25%) were chosen: The maximum bandwidth (B = 66%) was chosen due to larger selections generating QSA sets too similar in composition to regress fitted parameters on the central CTI values of the QSA sets. The minimum bandwidth (B = 25%) was chosen as a cutoff due to smaller B values containing too many QSA sets of fewer than 30 measurements selected to fit Eq. 1, which caused rapidly fluctuating parameterizations to occur.

Second, to analyse how varying central CTI values, which serve to inform climatology, influenced the fitted parameters in Eq. 1, QSA sets were generated for each bandwidth started from the bounds [0, B%] till [100%-B, 100%], with a step size of 1%. Each set was fit to Eq. 1, generating (100-B) QSA sets per bandwidth B. All QSA sets across all chosen B values had their resulting parameters ($m_1$, $m_2$, $s_1$, $s_2$), associated with their central CTI values, aggregated, totalling 224 sets of CTI-related parameterization results.

Finally, to mitigate the existence of noisy data impact on the regression of fitted parameters on the central CTI values of QSA sets, we applied a univariate smoothing algorithm (Appendix C) similar to that outlined in Cleveland et al. (1988) to smooth the fitted parameters and corresponding central CTI values of the QSA sets before performing the final regression analysis. This new composite set of smoothed QSA set parameters was then regressed on the associated QSA set central CTI values (Fig. 2c) to quantify the impact of climatological temperature on the model parameters. This provided four CTI-

dependent equations of the parameters ($m_1$, $m_2$, $s_1$, $s_2$) in Eq. 1, informing the climatological impacts on temperature tolerance and resilience of vegetative $\Phi_{PSIImax}$.

### 2.3.5 Comparison of CTI-informed and PFT-specific parameterizations of the temperature-$\Phi_{PSIImax}$ relationship

To assess the improvement of this CTI-informed parameterization for the temperature-$\Phi_{PSIImax}$ dynamic prediction, the cumulative sum of the prediction residuals from this CTI-informed parameterization was examined along the ascending order

of CTI values and compared with the counterparts from the PFT-specific parameterizations. The range of CTI over which the sum of residuals is reduced implies improved predictive power.

        To further identify the geographical regions within which the climatology's effect on the temperature-$\Phi_{PSIImax}$ relationship of a specific PFT was important to consider, we quantified the differences in estimated temperature resilience ($s_1$, $s_2$) and tolerance metrics ($T_{MC}$, $T_{MH}$) between CTI-informed and PFT-specific parameterizations at each 0.5° x 0.5° grid cell

across space. This comparison was constrained in the regions where CTI-informed parameterization showed improvement in prediction over the PFT-specific counterpart or could be compared with each other. Within this constrained domain, the CTI value at each grid cell was calculated using the 1985-2016 CRUNCEP v.7 hourly temperature dataset (Viovy et al., 2018) following the method introduced in Sect. 2.3.2. This was then applied to estimate CTI-based temperature resilience ($s_1$, $s_2$) and tolerance metrics ($T_{MC}$, $T_{MH}$) for the corresponding grid cell using the generated CTI-dependent $m_1$, $m_2$, $s_1$, and $s_2$ parameter

equations (Sect. 2.3.4). Covered PFTs at each grid cell were identified using the MODIS-derived present-day land cover data (Lawrence & Chase, 2007; Lawrence et al., 2016). To focus the study on PFTs that consistently reside within the CTI constrained domain, only PFTs whose total cover areas within the domain were at least 50% their global coverage were considered. PFT-specific temperature resilience ($s_1$, $s_2$) and tolerance metrics ($T_{MC}$, $T_{MH}$) at each PFT-covered grid cell were





estimated using PFT-specific parameterization (Sect. 2.2). The CTI-based parameters at each grid cell were finally compared

with the corresponding parameters from PFT-specific functions to identify the region at which climatological temperatures'

impact on the temperature tolerance and resilience of a specific PFTs' $\Phi_{PSIImax}$ values needs to be considered.

# 3 Results

## 3.1 Temperature response of $\Phi_{PSIImax}$ varies depending on PFT

Figure 3. PFT-specific parameterization results for 12 plant functional types, with the abbreviations (a) NET-Te, (b) NET-Bo, (c) BET-Tr, (d) BET-Te, (e) BDT-Tr, (f) BDT-Te, (g) BES, (h) BDS-Te, (i) C3-NAG, (j) C4-G, (k) C3-C, and (l) C4-C. The red line is the resulting PFT-specific model (Eq. 1). The blue dotted lines depict the slope of the model at the temperature $m_1$ ($m_2$) where $\Phi_{PSIImax}$ declines 50% from maximum $a$, and its slope is the resiliency parameter $s_1$ ($s_2$) for cold (hot) temperatures. The left (right) vertical green dotted lines depict the tolerance parameter $T_{MC}$ ($T_{MH}$) for cold (hot) temperatures, and the region

between them is the range of temperatures at which $\Phi_{PSIImax}$ remains constant.





Our results showed that the rectangular function (Eq.1) was able to capture the temperature dependence of $\Phi_{PSIImax}$ across all the gathered PAM fluorometry data. The cross-validation test resulted in statistically consistent $R^2$ of $0.49 \pm 0.03$ in both iterations (*p* value = 0.87). Data availability allowed for quality modelling of the PFT-specific temperature-$\Phi_{PSIImax}$ functions
for 12 plant functional types. Temperature variability explained more than 60% of $\Phi_{PSIImax}$ variations ($R^2 > 0.60$) for most of PFTs (Fig. 3), except for broadleaf evergreen temperate trees (BET-Te), C3 non-Arctic grasses (C3-NAG), and C4 grasses (C4-G) with $R^2$ values of 34%, 59%, and 46%, respectively (Fig. 3d,i,j). All PFTs followed the expected features of maintaining maximum $\Phi_{PSIImax}$ values of around 0.8 over a general temperature range from 16-34°C. Additionally, $\Phi_{PSIImax}$ significantly declined when temperatures get too hot (cold), depending on diverse temperature tolerances and resiliency of
different PFTs.

### 3.1.1 Tolerance

The $T_{MC}$ and $T_{MH}$ tolerance metrics of 12 PFTs varied from -0.7°C to 32.6°C and from 25.8°C to 42°C respectively, indicating large differences in cold and hot temperature tolerance of $\Phi_{PSIImax}$ values among 12 PFTs (Fig. 4a). Due to data gaps (Fig. 3c,e,l), the $T_{MC}$ of broadleaf evergreen tropical trees (BET-Tr) and broadleaf deciduous tropical trees (BDT-Tr), and the $T_{MH}$
of C4 crops (C4-C) were not used in mean and standard deviation (SD) calculations of $T_{MC}$ and $T_{MH}$. The mean and SD values of $T_{MC}$ and $T_{MH}$ for the rest of the PFTs are $16.3 \pm 12.2$°C and $33.8 \pm 4.8$°C, respectively. Among the 12 PFTs, the cold and hot tolerance responses of needleleaf evergreen temperate trees (NET-Te) are closest to the average, with values of 15.2°C and 33°C for $T_{MC}$ and $T_{MH}$, respectively (Fig. 4a). The $\Phi_{PSIImax}$ of C3 non-Arctic grasses (C3-NAG) showed the widest range of temperature tolerance to both cold and hot temperatures, ranging from -0.7°C to 34.5°C (Fig. 4a). Broadleaf evergreen shrubs
(BES) showed a similar range of tolerance temperatures, 3.5°C to 34.7°C (Fig. 4a). In contrast, needleleaf evergreen boreal trees (NET-Bo), broadleaf deciduous temperate trees (BDT-Te), C4 grasses (C4-G), and C3 crops (C3-C) showed weaker $\Phi_{PSIImax}$ tolerance to both hot and cold temperature compared to the average, ranging from 26.9-32.8°C, 26.4-30.8°C, 27.5-28.6°C, and 26.1-32.1°C respectively (Fig. 4a).

Most PFTs do not have simultaneous strong or weak tolerance to high and low temperature extremes, *i.e.*, high $T_{MH}$
values rarely occurred with low $T_{MC}$ values (strong temperature extreme tolerance), and low $T_{MH}$ values rarely occurred with high $T_{MC}$ values (weak temperature extreme tolerance). Broadleaf evergreen tropical trees (BET-Tr), broadleaf evergreen temperate trees (BET-Te), and broadleaf deciduous tropical trees (BDT-Tr) all had above-average hot tolerance, with $T_{MH}$ values of 36.9°C, 42°C, and 41.9°C, respectively (Fig. 4a). BDT-Tr and BET-Te were strongly hot tolerant PFTs, defined as $T_{MH}$ one standard deviation higher than the mean (Fig. 4a). The $T_{MC}$ value of BET-Te indicated a decline at temperatures below
32.6°C, although there was high variability in its $\Phi_{PSIImax}$ values at each temperature below 32.6°C. The declining trend of the $T_{MC}$ values of BET-Te gradually became obvious when temperatures fell below -7.7°C (Fig. 3d). BET-Tr (Fig. 3c) and BDT-Tr (Fig. 3e) lacked cold temperature measurements. Broadleaf deciduous temperate shrubs (BDS-Te) had a strong cold tolerance, with a $T_{MC}$ of 7.7°C, but did not have hot temperature tolerance ($T_{MH}$ = 26°C), unlike the other strong cold tolerance





PFTs of C3-NAG and BES (Fig. 4a). C4 crops (C4-C) were strong cold tolerant ($T_{MC}$ = -0.86°C), but due to low data

availability, the hot temperature responses of this group were unclear (Fig. 3l).

### 3.1.2 Resilience

The $s_1$ and $s_2$ resilience parameters of 12 PFTs varied from 1.56°C to 24.9°C and from 3.7°C to 12.9°C, respectively, indicating

that the resilience of $\Phi_{PSIImax}$ values to cold and hot temperature varied with PFT (Fig. 4b). A strong cold or hot resilience was

signalled by a large $s_1$ or $s_2$ value, and a slower decline of $\Phi_{PSIImax}$ as temperature changes beyond the ideal tolerance range.

Due to data gaps (Fig. 3c,e,l), the $s_1$ of broadleaf evergreen tropical trees (BET-Tr) and broadleaf deciduous tropical trees

(BDT-Tr), and the $s_2$ of C4 crops were not used in mean and SD calculations of $s_1$ and $s_2$. The mean and SD values of $s_1$ and

$s_2$ for the rest of PFTs were 11.6 ± 6.2°C and 7.1 ± 2.6°C, respectively. Among 12 PFTs, NET-Te had the closest-to-average

resilience to both low and high temperature ($s_1$ = 12.3°C; $s_2$ = 6.8°C) (Fig. 4b). The $\Phi_{PSIImax}$ values of the C3-NAG had the

least cold and second least hot temperature resilience, with an $s_1$ and $s_2$ of 3.9°C and 4.9°C, respectively (Fig. 4b). The next

least resilient PFT overall was BES, with an $s_1$ and $s_2$ of 8.99°C and 6.7°C, respectively (Fig. 4b). There was no PFT that had

both strong cold and strong hot temperature resilience, as defined by one SD more than the mean. The most temperature

resilient PFT overall was BDT-Te, with an $s_1$ and $s_2$ of 14.9°C and 12.99°C, respectively (Fig. 4b). Other PFTs that had higher

than average cold and hot resilience include C4-G ($s_1$ = 13.7°C; $s_2$ = 7.99°C) and C3-C ($s_1$ = 14.3°C; $s_2$ = 8.3°C) (Fig. 4b).

      PFTs that had strong cold resilience but weak hot resilience included NET-Bo ($s_1$ = 13.6°C; $s_2$ = 5.8°C), BET-Tr ($s_1$

= 20.9°C; $s_2$ = 6.6°C), BET-Te ($s_1$ = 24.9°C; $s_2$ = 4.0°C), and BDT-Tr ($s_1$ = 11.7°C; $s_2$ = 3.7°C) (Fig. 4b). BET-Te was a strong

cold resilience PFT. More cold temperature measurements for BET-Tr and BDT-Tr were needed to validate the current fitting

result (Fig 3c,e). PFTs that had strong hot resilience but weak cold resilience included BDS-Te ($s_1$ = 7.7°C; $s_2$ = 10.3°C) and

C4-C ($s_1$ = 1.56°C; $s_2$ = 12.9°C) (Fig. 3h,l). BDT-Te, BDS-Te, and C4-C were all strong hot resilient PFTs, although more hot

temperature observations for BDT-Te and C4-C are still needed for validating this fitting result.


### 3.1.3 Tolerance-resilience trade-off in PFT

Figure 4. Visualization of the PFT-specific tolerance and resilience parameters, including (a) the tolerance range of temperatures across which $\Phi_{PSIImax}$ remains consistently near the maximum modelled value (a) for the 12 modelled PFTs, as defined by the $T_{MC}$ and $T_{MH}$ values; (b) the cold ($s_1$) and hot ($s_2$) temperature resiliencies of the 12 modelled PFTs, labelled in the upper left and right corners respectively; (c) the trade-off between cold tolerance ($T_{MC}$) and resilience ($s_1$); (d) the trade-off between hot tolerance ($T_{MH}$) and resilience ($s_2$) for the 12 modelled PFTs. PFTs with a filled blue circle next to their parameter value in subplots (a-b) were not used in determining the respective cold temperature parameters' mean and standard deviation, and similarly PFTs with filled red circles next to their parameter values in subplots (a-b) were not used in determining the respective hot temperature parameters' mean and standard deviation. Open circles of red or blue in subplots (a-b) identify a PFT with data such that the resulting cold or hot parameter results should be taken with some caution. The gray lines in subplots (a-b) are the average parameter values. The high cold and high hot tolerance classifications are based on the PFT having a $T_{MC}$ or $T_{MH}$ value that is 1σ less than or greater than the respective average tolerance value and are represented by the blue and orange dashed lines respectively. The high resiliency parameters are defined both as having $s_1$ or $s_2$ that is 1σ greater than the respective average value across considered PFT values.





There was a positive correlation between $s_1$ parameters and $T_{MC}$ parameters, but negative correlation between $s_2$ parameters and $T_{MH}$ parameters for 12 PFTs (Fig. 4c,d). These results indicated a clear trend in temperature responses of the various PFTs' $\Phi_{PSIImax}$ values, in which the more temperature resilient PFTs were less temperature tolerant, and vice versa. The $s_1$ and $T_{MC}$ parameters of BET-Te (24.9°C, 32.6°C) indicated BET-Te was the most resilient but one of the least tolerant to cold temperature, whereas C4-C with a low $s_1$ (1.56°C) and the coldest $T_{MC}$ (-0.9°C) was extremely tolerant but one of the least

resilient to cold temperature (Fig. 4c). Besides C4-C and BDT-Te, which needed more high temperature observation for validating their fitted high resilience to hot temperature, BDS-Te was the third most resilient (10.3°C) but the least tolerant (26°C) to hot temperatures (Fig 4d). In contrast, C3-NAG showed both strong tolerance to cold temperature ($T_{MC}$ = -0.9°C) and high temperature ($T_{MH}$ = 38.9°C) but was the second least cold ($s_1$ = 3.85°C) and third least hot resilient ($s_2$ = 4.9) to temperature (Fig. 4c-d).

Based on this trade-off between temperature tolerance and resilience of the PFTs' $\Phi_{PSIImax}$ temperature response, we can classify 12 PFTs into three groups with different life history strategies of light partitioning in response to extreme temperatures. PFTs within the cold and hot temperature tolerant group (strongly temperature tolerant), including BES and C3-NAG, held the maximum $\Phi_{PSIImax}$ value (~ 0.8) within a wide (>20°C) temperature range. In contrast, PFTs within the cold and hot temperature resilient group (strongly temperature resilient), including BDT-Te, NET-Bo, C4-G, and C3-C, can still

keep higher $\Phi_{PSIImax}$ values in response to a large increase in hot temperature or decrease in cold temperature. NET-Te is a medium temperature tolerant and resilient PFT in between these extreme two groups. BET-Te, BDT-Tr, BET-Tr, C4-C, and BDS-Te are the temperature specialist group. They have a strong tolerance in either hot or cold temperatures (but not both) and are strongly resilient in opposite temperature extremes. Among them, C4-C and BDS-Te were strong cold tolerant and strong hot resilient PFTs, whereas BET-Tr and BET-Te were strong cold resilient and strong hot tolerant PFTs. BDT-Tr was

only tolerant to hot temperatures.

### 3.2 Climatology's influence on the temperature-$\Phi_{PSIImax}$ relationship

Here we first assessed the dependence of the temperature tolerance and resilience of $\Phi_{PSIImax}$ values on climatological temperature using results from ART ANOVA analyses (Sect. 3.2.1). Then we quantified the effect of climatological temperature on the temperature-$\Phi_{PSIImax}$ relationship by generating a climatological temperature index (CTI) using ART

ANOVA results and then parameterizing CTI effects on temperature tolerance and resilience parameters in the temperature-$\Phi_{PSIImax}$ function (Sect. 3.2.2). Finally, we assessed the performance of this CTI-informed parameterization of the temperature-$\Phi_{PSIImax}$ relationship by incorporating the field site sub-dataset to estimate and compared the sum of the absolute value of the prediction residuals between the CTI-informed parameterization and the PFT-informed parameterizations of the temperature-$\Phi_{PSIImax}$ relationship (Sect. 3.2.3).

### 3.2.1 Climatology temperature metrics are able to explain prediction errors





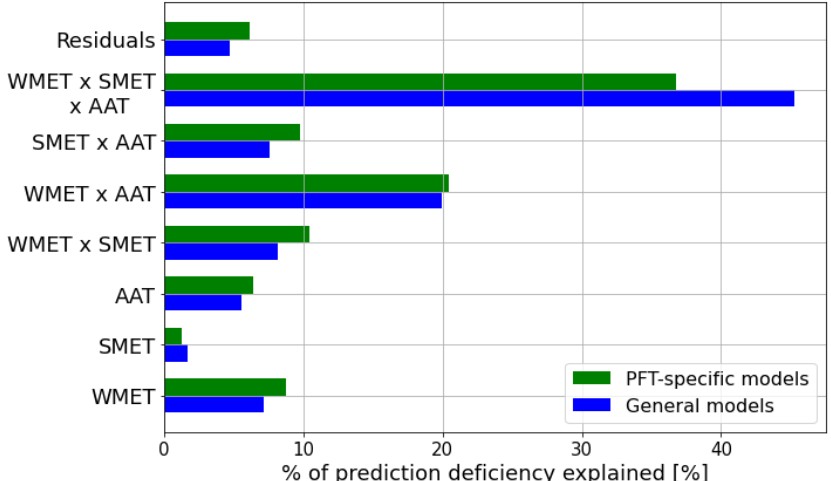

Figure 5. Comparison of ART ANOVA results between PFT-specific temperature-$\Phi_{PSIImax}$ functions and general temperature-$\Phi_{PSIImax}$ model residues using climatological temperature metrics. Here WMET and SMET refer to the median experienced temperature in the winter and summer respectively, while AAT refers to the annual average temperature.

Around 94% of variances in prediction residues by the PFT-specific temperature-$\Phi_{PSIImax}$ functions were able to be attributed between three climatological temperature metrics (WMET, SMET, and AAT) and their cross-terms. The interaction of WMET, SMET, and AAT showed the largest impact and explained around 37% of variations in prediction residues, followed by the cross effect of WMET and AAT with around 20% of variations in prediction residues associated with it (Fig. 5). In addition, the cross effect of SMET and WMET and the cross effect of SMET and AAT explained a similar amount (around 10%) of variations in prediction residues. Compared with the cross effect of three metrics, the main effects of individual metrics were relatively lower, with 8.78%, 6.42%, and 1.28% of variations in prediction residues associated with WMET, AAT, and SMET, respectively (Fig. 5).

There were consistent results between the ART ANOVA analysis for prediction residues estimated by the PFT-specific temperature-$\Phi_{PSIImax}$ functions and the general temperature-$\Phi_{PSIImax}$ function that resulted from fitting all data within the field site sub-dataset. WMET, SMET, and AAT as well as their interactions explained 96% of variances in prediction residues by the general temperature-$\Phi_{PSIImax}$ function (Fig. 5). This consistency indicated that besides PFT, climatological temperature played an important role in regulating the temperature-$\Phi_{PSIImax}$ relationship. The outsized difference in the amount of variance associated with the interaction and cross effects of three temperature metrics compared to the main effect of individual metrics suggested that annual mean temperature, together with the winter and summer temperature that the plants usually experienced in their habitats, determined the temperature response of plant photosynthetic light partitioning. Therefore, it was necessary to integrate three climatological temperature metrics to investigate the regulation of climatological temperature on the tolerance and resilience of $\Phi_{PSIImax}$ to climate change.





### 3.2.2 Climatology of plant habitats regulates the temperature tolerance and resilience of $\Phi_{PSIImax}$ values

The resulting regression of each parameter on CTI showed a strong correlation between CTI and each model parameter ($m_1$, $m_2$, $s_1$, $s_2$) in Eq. 1, in addition to the tolerance values ($T_{MC}$, $T_{MH}$). The strongest CTI dependence was with the hot temperature parameters ($m_2$, $R^2 = 0.94$; $s_2$, $R^2 = 0.87$) (Fig. 6b,d). Cold temperature parameters had slightly less dependence on CTI ($m_1$, $R^2 = 0.75$; $s_1$, $R^2 = 0.35$) (Fig. 6a,c). The CTI values were negatively correlated to the cold $T_{50}$ parameter $m_1$ (Fig. 6a), but positively correlated to the cold resilience parameter $s_1$ (Fig. 6c). In contrast, the CTI values were positively correlated to the

hot $T_{50}$ parameter $m_2$ (Fig. 6b) but negatively correlated to the hot resilience parameter $s_2$ (Fig. 6d). Built upon the regressions of $m_1$, $m_2$, $s_1$, and $s_2$ on CTI, the regression of $T_{MC}$ and $T_{MH}$ on CTI showed that $T_{MC}$ is negatively but not significantly correlated to CTI ($R^2 = 0.07$), whereas $T_{MH}$ was positively and significantly correlated ($R^2 = 0.93$) to CTI (Fig. 6e,f). These results indicated that the climate condition in the habitat in part affected the temperature tolerance and resilience of $\Phi_{PSIImax}$. Taking the averaged CTI of the field site sub-dataset as a reference, the $\Phi_{PSIImax}$ values of plants in the warmer than average habitats

(higher positive CTI) had a stronger tolerance to hot temperatures (Fig. 6f), at the cost of lower hot temperature resilience (Fig. 6d). Similarly, the $\Phi_{PSIImax}$ values of plants located in colder than average habitats (lower negative CTI) had a lower cold temperature resilience (Fig. 6c), but no statistically significant change in cold temperature tolerance with CTI (Fig. 6e).

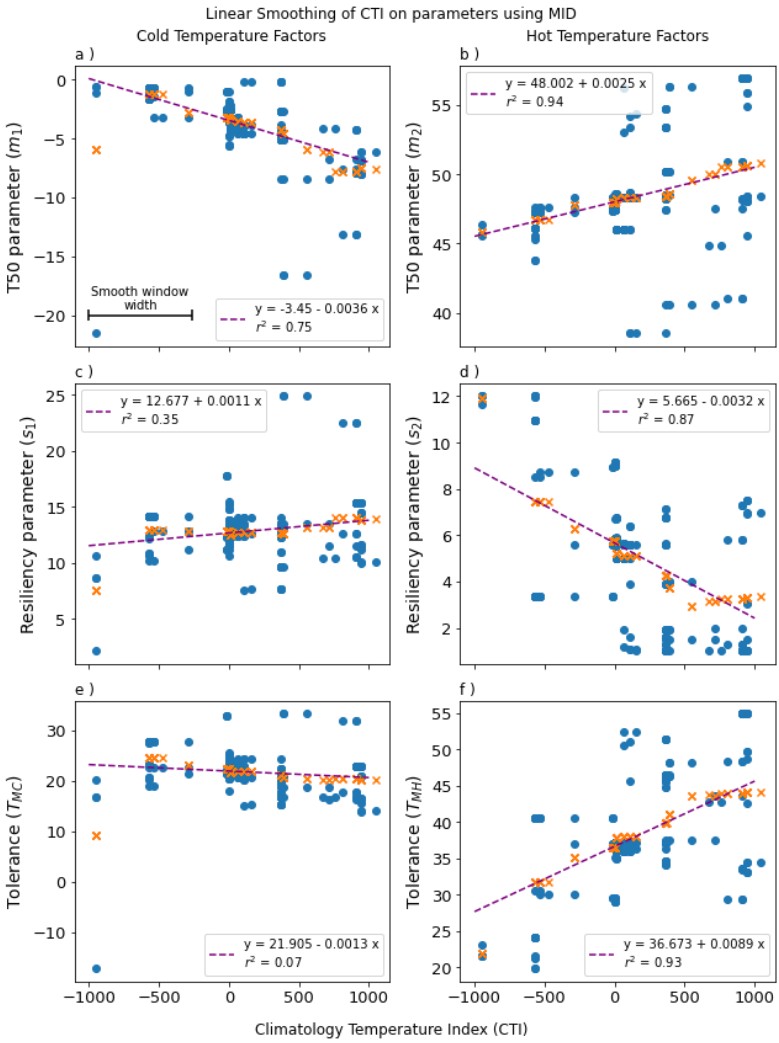

Figure 6. Linear Smoothed MID regression of temperature resilience and tolerance parameters on the climatological
temperature index (CTI) using the QSA approach: (a) $m_1$-CTI, (b) $s_1$-CTI, (c) $m_2$-CTI, & (d) $s_2$-CTI. Here m1, m2, S1, S2 refer
to the original 4 parameters in Eq. 1. The original estimation of each parameter and the corresponding mean CTI of a QSA set
are displayed in blue-filled circles. The smooth window width featured in (a) the top left plot is centered upon each QSA result,
and all values within that CTI range are averaged. The orange cross symbols (x) are the resulting smoothed values, which were
then run through a linear least square regression. The resulting regressions are displayed as purple lines.

**3.2.3 Improved predictions of the temperature-Φ$_{PSIImax}$ dynamic using CTI-informed parameterization**

By substituting observation temperature and site-specific CTI within the field site sub-dataset into CTI-informed and PFT-
specific parameterizations, we showed that utilizing the CTI-informed parameterization improved the accuracy of Φ$_{PSIImax}$
prediction by 4.3% on average, but this improvement was achieved across the range $150 \leq |CTI| \leq 900$ (Fig. 7). This range
represents the transition away from the mean state of SMET, WMET, and AAT for the field site sub-dataset. For |CTI| below
150, the improvement in the sum of residuals was not significant, implying that CTI-informed temperature dependence of





$\Phi_{PSIImax}$ agreed with the PFT-specific results in aggregate. Our results highlighted that climatology's effect on temperature tolerance and resilience of $\Phi_{PSIImax}$ values needs to be considered in regions with $150 \leq |CTI| \leq 900$.

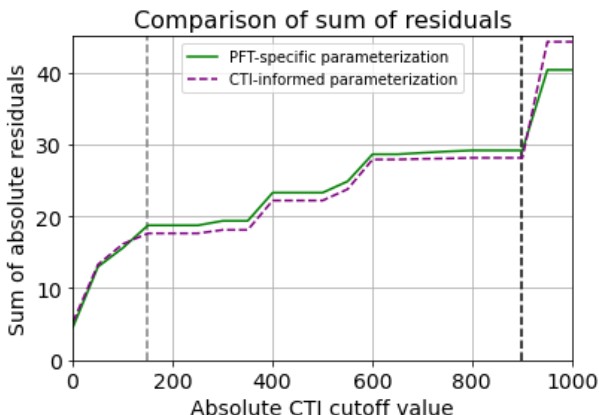

Figure 7. Comparing the sum of the magnitude of the prediction residuals between CTI-informed and PFT-specific parameterization of vegetative $\Phi_{PSIImax}$-temperature relationship.

### 3.3 Global distribution of CTI and its implications

To better identify the regions within which the impact of climatological temperatures on the temperature tolerance and resilience of $\Phi_{PSIImax}$ values needs to be considered in predicting specific PFT $\Phi_{PSIImax}$ values, we estimated the global distribution of the CTI values at each 0.5° x 0.5° grid cell within the region of $|CTI| \leq 900$ (Sect. 3.3.1). Based on this distribution, spatial distributions of the CTI-informed parameters ($m_1$, $m_2$, $s_1$, $s_2$) and tolerance metrics ($T_{MC}$, $T_{MH}$) were calculated (Sect. 3.3.2). A final assessment of the spatial pattern of differences between the CTI-informed parameters and their PFT-specific counterparts was performed on the regions that contained said PFT coverage (Sect. 3.3.3).

### 3.3.1 CTI global pattern

The global distribution of CTI showed a clear latitude gradient. A large area of the mid-latitudes had CTI values close to zero (Fig. 8a), indicating these regions had very similar CTI indices to the mean state, which referred to the mean of all the respective CTI values within the field site sub-dataset. The CTI values trended to be more positive from the middle latitude to the tropical region and more negative from the middle latitude to the polar regions (Fig. 8a).

The following comparison between CTI-informed and PFT-specific parameterization of temperature-$\Phi_{PSIImax}$ relationship was constrained to the land grid cell with $|CTI| \leq 900$, where the CTI-informed parameterization can be improved ($150 \leq |CTI| \leq 900$) or hold comparable ($|CTI| < 150$) in the predictive power of temperature-$\Phi_{PSIImax}$ relationship, compared with the corresponding prediction by PFT-specific parameterization. The land area with $|CTI| \leq 900$ accounted for ~53% of the earth's land area, which was distributed in the latitudinal bands of 60° S - 80° N, especially in the latitudinal bands of 30°S-60° S and 30°N-70°N, where almost 100% of land grid cells at the same latitude had $|CTI| \leq 900$ (Fig. 8b). Spatially, these





regions included much of South America outside of the Amazon, Africa, and most of Canada and Russia (Fig. 8c). The upper

Andes and Mexico's Sierra Madre ranges, a large region in Ethiopia, and much of Libya had CTI that is much lower than the

surrounding regions at the same latitude, falling with the CTI bounds of improved or comparable predictive power. Notable

regions that fell outside the bounds were both the Amazon and Indonesian rainforests, as well as the Indian subcontinent and

much of the Congo rainforest (Fig. 8c).

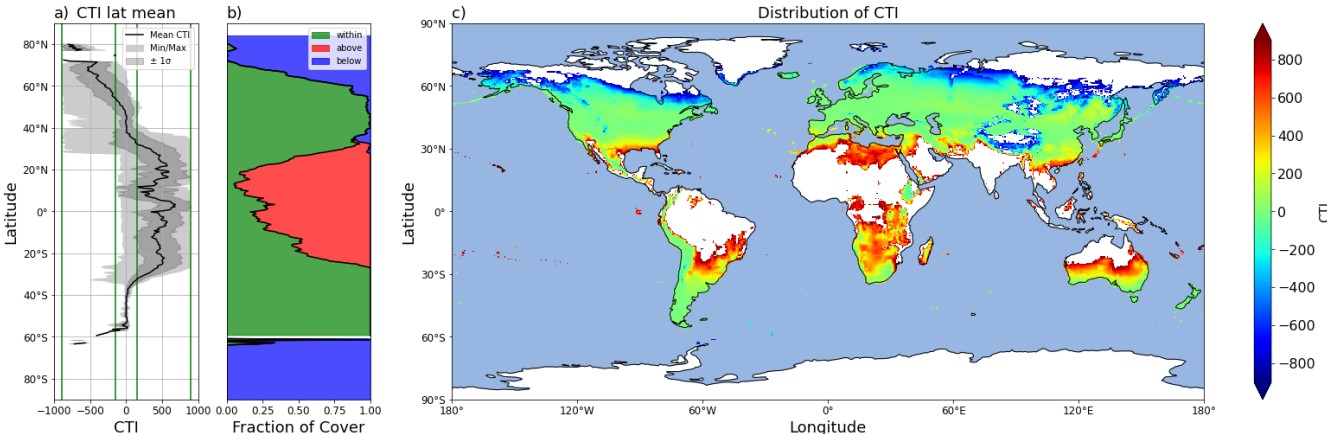

Figure 8. Global Distribution of CTI values and their latitudinal trends. (a) Latitude mean CTI values of all land grid cells with |CTI| not larger than |900| (black line), with both the minimum and maximum (light gray), as well as ± the standard deviation (σ) (dark gray). Four green lines from the left to the right demonstrate the CTI bounds of -900, -150, 150, and 900, respectively. The range of 150 ≤|CTI| ≤900 refers to the CTI range at which the temperature-$\Phi_{PSIImax}$ relationship was improved by CTI-informed parameterization compared with the PFT-specific parameterization. (b) The percentage of land grid cells at each 530 latitude that were below (blue), within (green), and above (red) the range of |CTI| ≤ 900. (c) Map depicting the global distribution of grid cells at which |CTI| ≤ 900.

### 3.3.2 Latitudinal variation in CTI-informed parameters of temperature regulation on $\Phi_{PSIImax}$

Using the CTI-informed parameterizations, the cold/warm $T_{50}$ parameters ($m_1$, $m_2$), temperature resilience parameters ($s_1$, $s_2$) and resultant temperature tolerance parameters ($T_{MC}$, $T_{MH}$) of vegetative $\Phi_{PSIImax}$ values showed clear latitudinal gradients

across the globe. The latitudinal mean $m_1$ increased from -5.5°C to -3.5°C across the transient band moving from tropical to

mid-latitudes (30-40 °N & °S) and from -3.5°C to almost 0°C from mid-latitudes to high latitudes (50-80°N & 55-65°S), but

held values around -5.5°C in the tropics (30°N to 30°S) and around -3.5°C in the mid-latitudes (40-50 °N & 40-60 °S) (Fig.

9a). Parameter $s_1$ also showed less latitudinal variability around 30°N to 30°S and the mid-latitude band around 40-50 °N &

40-60 °S, but an inverse trend compared with the latitudinal variability of $m_1$, decreasing from 13.2 in tropical (30°N to 30°S)

to 12.7 in mid-latitudes (30-40 °N & °S) and from 12.7 in mid-latitudes to 11.6 in high latitudes (50-80°N & 55-65°S) (Fig.

9b). The cold tolerance $T_{MC}$, being a linear combination of $m_1$ and $s_1$, exhibited a similar trend as $m_1$, with almost 2°C of

variation across latitudes which is smaller in comparison with the latitudinal variability of $m_1$ (Fig. 9c). These results indicated

the $\Phi_{PSIImax}$ in temperate habitats across 40°-50 °N & 40-60 °S had similar cold temperature $T_{50}$ and cold temperature resilience



and tolerance. However, $\Phi_{PSIImax}$ tended to have a higher cold temperature $T_{50}$ and be less tolerant and resilient to cold
temperatures from 30°-40° N & S and from 50°-80° N and from 55°-65° S.

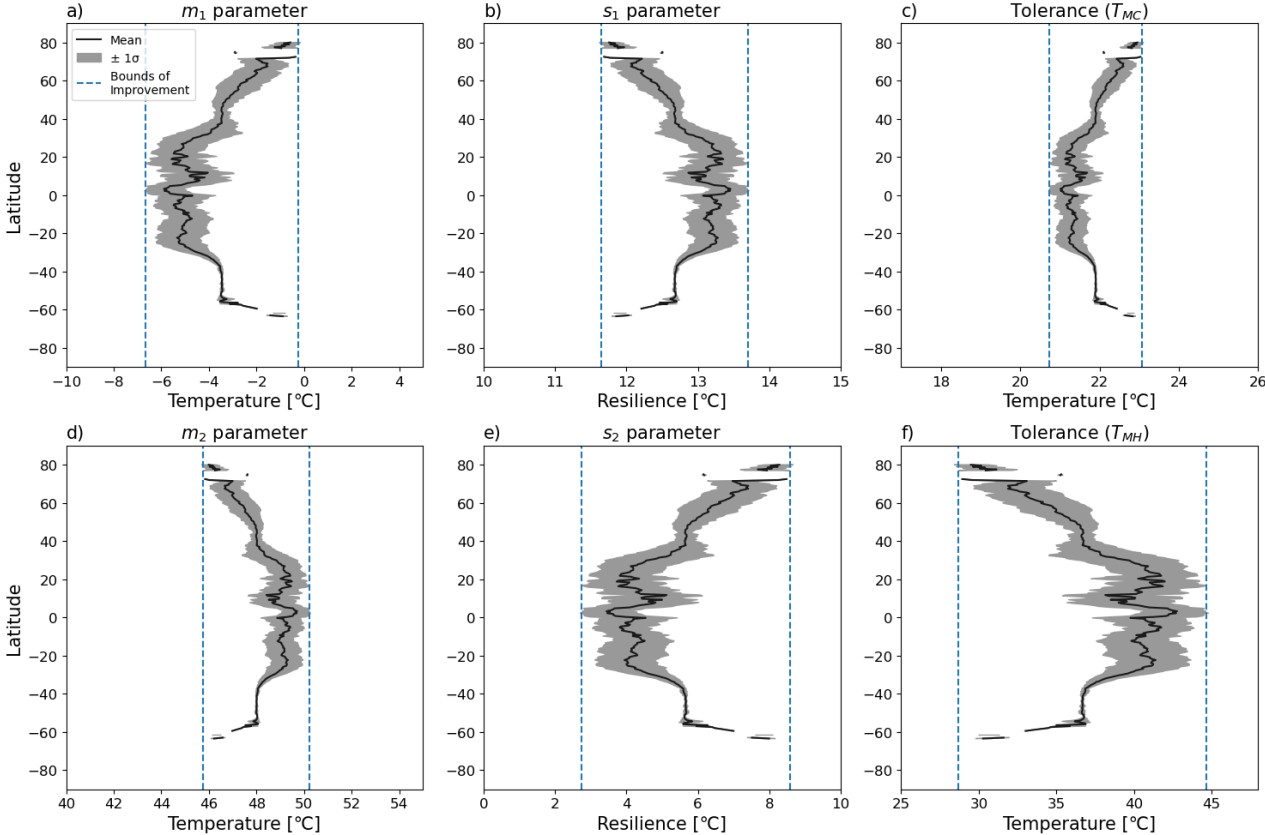

Figure 9. Latitudinal mean and standard deviation (σ) of CTI-informed model parameters, including (a-b) cold and hot $T_{50}$ parameters ($m_1$, $m_2$) at which $\Phi_{PSIImax}$ values decrease by 50%, (c-d) cold and hot temperature resilience parameters ($s_1$, $s_2$) in Eq. 1, and (e-f) derived cold and hot temperature tolerance metrics ($T_{MC}$, $T_{MH}$) for land grid cells with the CTI value <|900|.

The latitudinal mean $m_2$ decreased from 49.2°C to 47.9°C from 20°-40° N & S and from 47.9°C to 45.8°C from middle to high latitude (50°-80° N, 55°-65° S), with a maximum value of 49.7° across the tropics, and an almost constant value of 47.9°C across 40°-50° N & S (Fig. 9d). Similar to the relationship between the latitudinal trend of $m_1$ and $s_1$, the latitudinal mean $s_2$ increased across latitudes where the $m_2$ decreased, with a greater latitudinal variation compared to $s_1$ (Fig. 9e). Similar to $m_1$ and $s_1$, $m_2$ and $s_2$ also had less variation in the tropic (30°N to 30°S) and the mid-latitude band around 40-50 °N and 40-60° S. The hot tolerance $T_{MH}$, being a linear combination of $m_2$ and $s_2$, exhibited a similar trend to $m_2$, with around 15°C of variation across latitudes being larger in comparison with the latitudinal variability of $m_2$ (Fig 9g). These findings also suggested that the vegetation at temperate habitats across 40-50° N & S had a similar response of $\Phi_{PSIImax}$ value to hot





temperature. In contrast, the $\Phi_{PSIImax}$ values tended to be less tolerant but more resilient to hot temperature across the tropical-
to-middle latitude transition regions (30°-40° N & S) and middle-to-high latitude transition regions (50°-80° N, 55°-65° S).

Corresponding to the latitudinal pattern of $T_{50}$, resilience, and tolerance parameters (Fig. 9) with the spatial pattern of
CTI values (Fig. 8), $\Phi_{PSIImax}$ values became less cold tolerant and resilient, and less hot tolerant but more hot-resilient along
warm-to-cold climatological gradients (CTI gradients).

### 3.3.3 Spatial distribution of the differences between CTI-informed and PFT-specific parameterizations

To identify specific geographical locations where PFT-specific temperature tolerance or resilience of $\Phi_{PSIImax}$ differed from
the CTI-informed counterparts, we calculated the difference in $T_{MC}$, $T_{MH}$, $s_1$, and $s_2$ between CTI-informed and PFT-specific
parameterizations at each grid cell. Only PFTs with total cover area within the grid cells of $|CTI| \leq 900$ accounting for $\geq 50\%$
of its global total distribution area were included in this analysis, resulting in the following PFTs: NET-Te, NET-Bo, BET-Te,
BDT-Te, BES, BDS-Te, C3-NAG, and C3-C. Latitudinal mean and standard deviation of compared results for all grid cells
that contained said PFT were calculated and shown in Fig. 10.

Our results showed that PFT-specific $T_{MC}$ for all analysed PFTs had larger differences compared with the CTI-
informed counterparts, but no obvious latitudinal variability was observed, as the magnitude of the variation was about 2°C.
CTI-informed $T_{MC}$ for BET-Te, NET-Bo, BDT-Te, and C3-C had a value below the corresponding PFT-specific $T_{MC}$ (stronger
cold tolerance) by 11°C, 5°C, 5°C, and 3°C, respectively, while CTI-informed $T_{MC}$ for NET-Te, BDS-Te, BES, and C3-NAG
had values above the corresponding PFT-specific $T_{MC}$ (weaker cold tolerance) by around 6-7°C, 14 °C, 18°C, and 22-23°C,
respectively (Fig. 10a). Similarly, the differences between CTI-informed and PFT-specific cold resilience metric $s_1$ showed
slight latitudinal variability but differed among different PFTs. The CTI-informed $s_1$ for C3-NAG was two times greater than
its PFT-specific counterpart (210-250%), followed by BDS-Te and BES with around 40-60% greater CTI-informed $s_1$ than
PFT-specific counterpart (Fig. 10b), indicating that CTI-informed $\Phi_{PSIImax}$ values of C3-NAG, BDS-Te, and BES were more
resilient to cold temperature than the corresponding PFT-specific estimation. In contrast, CTI-informed $\Phi_{PSIImax}$ values of C3-
C, NET-Bo, BDT-Te, and BET-Te became less resilient to cold temperatures than PFT-specific parameterization, with the
$\Phi_{PSIImax}$ value of BET-Te having the largest cold resilience reduction (50%, Fig. 10b). Compared with the PFTs discussed
above, there was almost no significant difference in NET-Te between the two parameterizations. Overall, our results indicated
that C3-NAG, BES, BDS-Te, and BET-Te were key PFTs with larger changes in cold tolerance (>10 °C) and cold resilience
(>40%) of CTI-informed $\Phi_{PSIImax}$ values compared with the corresponding PFT-specific counterparts.



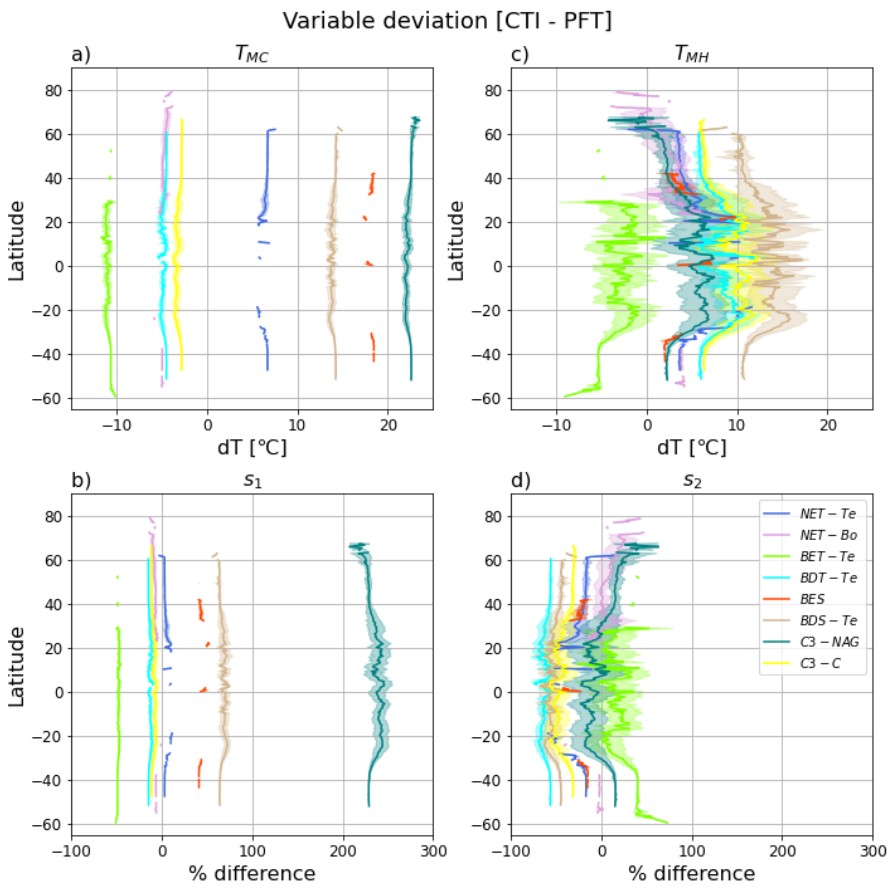

Figure 10. Latitudinal mean and standard deviation (σ) of the difference between CTI-informed and PFT-specific temperature tolerances metrics and resilience parameters of vegetative $\Phi_{PSIImax}$ values, including (a) cold temperature tolerance ($T_{MC}$), (b) cold temperature resilience ($s_1$), (c) hot temperature tolerance ($T_{MH}$), and (d) hot temperature resilience ($s_2$). Here only grid
cells having |CTI| values ≤ 900 and targeted PFT cover are included in the calculation of latitudinal mean and standard deviation. Targeted PFTs refer to PFTs which total covered area within the grid cells having |CTI| ≤ 900 accounts for ≥ 50% of its global total distribution area. Each PFT-specific result is shown as separate colors, with the shaded regions of the same color are ±1σ at each latitude.

The differences in $T_{MH}$ and $s_2$ between CTI-informed and PFT-specific parameterizations for all analysed PFTs
showed larger latitudinal variability compared to their cold temperature counterparts. CTI-informed $T_{MH}$ for BET-Te was lower (lower hot tolerance) than the PFT-specific $T_{MH}$ by around 2.5 ~ 10°C with a larger reduction in $T_{MH}$ observed in southern middle latitudes (Fig. 10c). In contrast, CTI-informed $T_{MH}$ for BDT-Te, BES, BDS-Te, and C3-C all larger than the PFT-specific counterparts, with the increased $T_{MH}$ difference between the two parameterizations from high to lower latitude, especially from 25°-40° N & S (Fig. 10c). Among these PFTs, CTI-informed $T_{MH}$ for BDS-Te showed the largest increase
with the latitudinal mean values by around 10 - 17°C across 60°N-60°S, compared with the PFT-specific counterpart. CTI-informed latitudinal mean $T_{MH}$ for C3-C and BDT-Te increased by 6 ~ 12°C across the covered latitude region, compared with





the PFT-specific counterpart. Unlike PFTs discussed above, NET-Te, C3-NAG, and NET-Bo had diverging $T_{MH}$ differences between the two parameterizations across latitudinal gradients. Compared with the PFT-specific counterpart, CTI-informed $T_{MH}$ for NET-Te showed a maximum 2.5°C of decrease around 60°N, but a maximum 10°C increase around 20°N (Fig. 10c).

Similarly, CTI-informed $T_{MH}$ for C3-NAG and NET-Bo decreased by 5°C beyond 60°N but increased by around 1 ~ 7°C across 60°N-50°S, compared with the corresponding estimation by PFT-specific parameterization (Fig. 10c).

The CTI-informed $s_2$ indicated that the $\Phi_{PSIImax}$ value of BET-Te should have higher hot resilience than the PFT-specific counterparts on average across all latitude bands, with a wide range of percentage increase (7- 72%) across 30°N-60°S. In contrast, BDT-Te, BDS-Te, C3-C, NET-Te, and BES all had lower hot resilience of $\Phi_{PSIImax}$ than the PFT-specific

estimation (Fig. 10d). CTI-informed hot resilience of $\Phi_{PSIImax}$ for BDT-Te and BDS-Te decreased by around 45% - 68% from 60°N-50°S, followed by C3-C, BES, and NET-Te with around 17% - 52% decrease in CTI-informed hot resilience of $\Phi_{PSIImax}$ compared to their PFT-specific counterparts (Fig. 10d). CTI-informed $s_2$ for NET-Bo and C3-NAG both had diverging differences with the corresponding PFT-specific counterparts across latitudinal gradients. In the northern latitudes, the hot temperature resilience parameter $T_{MH}$ for C3-NAG increased by up to 25% from 30°N to 60°N and from 30°S to 60°S, spiking

to over 50% beyond 60°N, but decreased from 30°N to 30°S, compared with the corresponding PFT-specific counterparts. NET-Bo showed no significant difference between CTI-informed and PFT-specific hot resilience estimation in the southern latitudes and around 30°N-50°N, while showed up to 50% increase in CTI-informed values in around 50°N-80°N than PFT-specific counterparts. In total, these findings indicated that BDS-Te, BET-Te, BDT-Te, and C3-C were PFTs with larger changes in hot tolerance (>5°C) and hot resilience (>10%) of CTI-informed $\Phi_{PSIImax}$ values compared with the corresponding

PFT-specific counterparts. However, the differences in these hot temperature response metrics between the two parameterizations showed large variability across the latitude.

## 4 Discussion

### 4.1 PFT variation in the temperature tolerance and resilience of photochemical efficiency

Our study gathered global-scale PAM fluorometry observations to quantify temperature regulation on $\Phi_{PSIImax}$ values of 12

PFTs. The developed PFT-specific rectangular functions can capture the optimal $\Phi_{PSIImax}$ value of 0.8 within different ranges of temperature and the decrease of $\Phi_{PSIImax}$ in each PFT with colder and hotter temperatures (Fig. 3). The variability of $\Phi_{PSIImax}$ around the ideal temperature range echoed observed variability across species in response to similar conditions (Li et al., 2004). Few previous studies (Sastry and Barua, 2017; Slot et al., 2019; Perez & Feeley, 2020; Tiwari et al., 2021; Kunert et al., 2021) utilized different functions to fit this heat response of $\Phi_{PSIImax}$ values to the warming climate. These studies mainly focused on

specific species, such as tropical evergreen and deciduous species. To our knowledge, our study is the first global-scale effort that quantifies the differences in this temperature-$\Phi_{PSIImax}$ relationship among 12 general PFTs. The generated PFT-specific temperature-$\Phi_{PSIImax}$ functions can be directly applied to adjust the $\Phi_{PSIImax}$ parameter for simulating temperature feedback of photosynthetic energy partitioning in terrestrial ecosystem models (TEMs) and ESMs.





One advantage of the developed temperature-$\Phi_{PSIImax}$ model is its capability for quantifying the temperature tolerance

and resilience of $\Phi_{PSIImax}$ values and assessing their differences among different PFTs. Our results highlighted the trade-off
scheme of temperature responses of the various PFTs' $\Phi_{PSIImax}$ values, in which the more temperature-resilient PFTs were less
temperature tolerant, and vice versa (Fig. 4). Our finding was consistent with previous studies (Tiwari et al., 2021; Kunert et
al., 2021) that also found a negative relationship between temperature tolerance and resilience metrics for some tropical tree
species, although the definition of temperature tolerance and resilience metrics in Tiwari et al. (2021)'s study was not

completely consistent with our definition. This trade-off may be associated with the selection of different PFTs' protection
strategies under current and historical temperature stress. Temperature-resilient PFTs may protect PSII from damage via NPQ
mechanisms, such as heat shock proteins (HSP) and the xanthophyll cycle (Adams & Demmig-Adams, 1994; Wahid et al.,
2007). Temperature tolerant PFTs may have more flexibility when it comes to PSII protections, either due to phenological
responses (i.e., leaf senescence under cold/hot stress, maximizing warm season growth (Sakai, 1981)), or physiological

feedback (i.e., evapotranspiration cooling under hot stress (Havaux, 1992)).

According to this trade-off between temperature tolerance and resilience of the PFTs' $\Phi_{PSIImax}$ temperature response,
we identified the cold and hot temperature tolerant PFT group (BES and C3-NAG), the cold and hot temperature resilient PFT
group (BDT-Te, NET-Bo, C4-G, C3-C), and the temperature specialists PFT group (BET-Te, BDT-Tr, BET-Tr, C4-C, and
BDS-Te). NET-Te adopted a generalist strategy, with tolerance and resilience values near the mean across PFTs, though this

may be inherent bias in the datasets. The higher tolerance of C3-NAG to both cold and hot temperatures was probably
associated with its discrete distribution across diverse habitats, such as Amazon and boreal regions (Ke et al., 2012). Long-
term exposure to diverse temperatures had evolved diverse C3-NAG species and cultivars, which had distinct abilities to
suppress oxidative stress under temperature stress and therefore broad heat/cold tolerance of $\Phi_{PSIImax}$ (Soliman et al., 2012;
Filho et al., 2018). Similarly, BES, which was usually distributed in the Mediterranean climate with typical cold winter and

drought summer (Ke et al., 2012), had been found to maintain higher photochemical yields under hot and cold temperatures
by adjusting vegetation structure and decreasing chlorophyll content (Oliveira, 2000). BDT-Te was the most heat resilient
PFT, reflecting a strong heat shock protein (HSP) and stomatal conductance response (Solhaug & Haugen, 1998; Wittmann &
Pfanz, 2007; Song et al, 2014). Some species within BDT-Te had a phenotypic abscission response to high temperature (Shirke
& Pathre, 2003), which may be a reason behind the mild variability in $\Phi_{PSIImax}$ measurements at high temperature (Fig. 3f).

The higher cold and hot temperature resilience of C3-C may be related to crop engineering selection of crop genotypes that
can be more resilient to extreme temperature and weather events (Basu et al., 2009*; Molina-Bravo et al., 2011; Zhou et al.,
2015; Sharma et al., 2017). In the temperature specialists PFT group, BET-Te was the strongest hot tolerant and strongest cold
resilient PFT, which may be due to its adaptation to hot temperatures by having a phenological mechanism with year-round
leaf turn-over (Williams-Linera, 1997), and having freezing resistance in leaves (Sakai, 1981). Similar to BET-Te, BDT-Tr

could maintain higher $\Phi_{PSIImax}$ at temperatures up to 42°C. The high heat tolerance of BDT-Tr was consistent with Tiwari et
al. (2021) and was probably associated with the synthesis of HSPs under long-term exposure to topical high temperature
(Taleisnik & Grunberg, 1994).





Our definitions of temperature tolerance and resilience are not completely the same as previous studies, however, they can be comparable after conversion. For example, previous studies (Tiwari et al., 2021; Kunert et al., 2021) estimated $T_{50}$

as the same definition of $m_2$ in our PFT-specific temperature-$\Phi_{PSIImax}$ model (Eq.1), but quantified temperature resilience of $\Phi_{PSIImax}$ as decline width (DW = $T_{95} - T_5$) and temperature tolerance of $\Phi_{PSIImax}$ as $T_5$. Here $T_5$ and $T_{95}$ referred to the temperature at which $\Phi_{PSIImax}$ declined with temperature change by 5% and 95%. By calculating $T_{50}$, $T_{95}$ and $T_5$ using the fitted Eq. 1, we found that the parameterization for BET-Tr using our model resulted in $T_5$ of 42.35°C, $T_{50}$ of 49.98°C, and a resilience metric ($T_{95} - T_5$) of 15.26°C, respectively. This result was close to Tiwari et al. (2021) estimations of dry (wet) season $T_5 = 43.5$°C

(41.9°C), $T_{50} = 51.6$°C (49.4°C), and $T_{95}$ - $T_5 = 16.6$°C (15.4°C) averaged across diverse BET-Tr species. Also, our estimation of $T_5$, $T_{50}$, and $T_{95}$ values for NET-Te were 38.7°C, 46.7°C, and 54.7°C, respectively. The estimated $T_5$ and $T_{95}$ fell within the Kunert et al. (2021) estimated span of the 6 NET-Te species' for $T_5$ (38.5- 43.1°C) and $T_{95}$ (53.9-57.5°C). However, the estimated $T_{50}$ was somewhat lower than the corresponding estimation of 47.8 - 52.3°C by Kunert et al. (2021).

### 4.2 Climatology-driven convergent temperature-$\Phi_{PSIImax}$ response across PFTs in a certain region

The formation and application of the CTI as an integrated indicator of climatological annual mean temperature, winter minimum temperature, and summer maximum temperature, allowed for quantifying the variation in temperature tolerance and resilience of photochemical efficiency with climatological gradients. This CTI-incorporated temperature-$\Phi_{PSIImax}$ parameterization (Fig. 7) was comparable with the corresponding PFT-informed temperature-$\Phi_{PSIImax}$ parameterization within the region of |CTI| < 150 and increased predictive power within the region with $150 \leq |CTI| \leq 900$, implying that the irreversible

variability of photochemical efficiency of a plant with temperature change mainly depends on its acclimation and adaptation to climatology in these regions. PFT has been widely applied to interpret the variation among plants in physiological, morphological, and phenological traits and correlated to plant adaptation to local environmental conditions and to plant resource capture and survival strategies (Reich et al., 2003; Kelly et al., 2021). However, our results demonstrated that the temperature sensitivity and tolerances of photochemical efficiency trait in a certain CTI band (CTI ≤ |900|) is convergent and

can be parameterized using a CTI-informed fundamental function. This universal function can be directly incorporated into TEMs and ESMs to parameterize the temperature feedback of photosynthetic light reactions, instead of using PFT-specific parameterization which requires more parameters. Our finding was similar to the study of Heskel et. al. (2016) that found consistent temperature sensitivity of leaf respiration across several PFTs and biomes and parameterized it using a universal temperature-dependent function. The convergent temperature responses of these plant traits may reflect occurrence of

phenotypic plasticity or ecotypic variations as a result of plant acclimation and adaptation to its inhabited climatological temperature (Marias et al., 2016) or universal metabolic constraints on vegetation temperature feedback (Heskel et al., 2016).

The global distribution of the regions within the bounds of improved prediction ($|150| \leq CTI \leq |900|$) were concentrated around subtropical regions (30°-40° N & S) and along the transition zones from the mid-latitudes to the polar regions (50°N-70°N and 50°S-60°S) (Fig. 8a, c). These regions typically had a large inter-seasonal variability in temperature

or contrasting precipitation seasonality coupling with hot temperature through the year. Several observations (Wahid et al.,




2007; Soliman et al., 2012; Marias et al., 2016) had shown that plant species in a variable climatological environment has been
more phenotypically plastic and representative "experienced temperature" through different feedstock, such as synthesis of
non-structural carbon under temperature stress, generation of heat shock proteins (HSP) to avoid heat damage and increasing
evaporative cooling through adjusting stomatal size and density. Moreover, moisture stress may modulate plant plastic capacity
due in part to an attempt to maximize water use efficiency (WUE) and maintain homeostasis (Wahid et al., 2007; Lin et al.,
2015; Marias et al., 2016). The improved prediction of temperature-$\Phi_{PSIImax}$ relationship in these regions by our CTI-informed
parameterization was consistent with previous observation in the subtropical region that species from contrasting climate of
origin (desert vs. coastal) didn't show significantly different tolerance when growing in the common environment (Knight and
Ackerly, 2001).

**4.3 The correlation of temperature tolerance and resilience of photochemical efficiency with its local habitat climatology**

Our results indicated that $\Phi_{PSIImax}$ tends to be less cold tolerant and resilient, but less hot tolerant and more hot-resilient along
warm-to-cold climatological gradients, except for the temperature region across 40-50°N & 40-60°S. The decrease in both
cold tolerance and resilience of photochemical efficiency along warm-to-cold climatology was probably because that more
extreme and highly frequent cold temperature (e.g., chilling) in colder regions may damage metabolic processes and inhibit
adaptive vegetation feedback, such as sugar synthesis, osmotic production, and pigment synthesis (Hajihashemi et al., 2018),
or induce initiation of NPQ to dissipate the resulting excess energy (Rapacz et al., 2004). In contrast, our finding of decrease
in hot tolerance along warm-to-cold climatology supported previous conclusions that vegetation distributed in warmer climates
have greater heat tolerance (Smillie & Nott, 1979; Salvucci & Crafts-Brander 2004; Marias et al., 2016, Fadrique et al., 2022).
Lack of a latitudinal trend of temperature tolerance and resilience across 40-50°N & 40-60°S was probably due to the highly
variable climatology in the mid-latitudes. Therefore, the temperature tolerance and resilience of $\Phi_{PSIImax}$ may reflect local
climate properties.

**4.4 The advantage of CTI-informed parameterization over PFT-specific parameterization is PFT-specific**

Our results suggested that the importance of climatological regulation on temperature-$\Phi_{PSIImax}$ relationships differs among
different PFTs. In the region with comparable or improved prediction power by CTI-informed parameterization, CTI-informed
$\Phi_{PSIImax}$ values of C3-NAG, BES, BDS-Te, and BET-Te showed larger changes in cold tolerance (> 10°C) and cold resilience
(>40%) compared with the corresponding PFT-specific counterparts, whereas CTI-informed $\Phi_{PSIImax}$ values of BDS-Te, BET-
Te, BDT-Te, and C3-C had larger changes in hot tolerance (>5°C) and hot resilience (>10%) compared with the corresponding
PFT-specific counterparts. As discussed in Sect. 4.2, this result may reflect the acclimation and adaptation of these PFTs to
local temperature variability, instead of the PFT-unified life history strategy (Curtis et al., 2016). For example, C3-NAG and
BES are distributed across wide latitude ranges, 67.5°N-52°S and 52°N-43.5°S respectively, with diverse climatological
variability in distribution space, whereas BDS-Te and BET-Te were mainly distributed in temperate regions with a large
seasonal variability in temperature (Lawrence & Chase, 2007; Ke et al., 2012; Lawrence et al., 2019).



## 4.5 Uncertainty and future work

Fully identifying the underlying mechanisms of why $\Phi_{PSIImax}$ declines with temperature is beyond the scope of this study,
however they likely fall into two categories: photophysical effects and biophysiochemical effects. From the photophysical
perspective, as temperature changes outside of an idealized range, sustained NPQ is known to increase in some plants (Porcar-
Castell, 2011), though the variation of experimental method across the gathered studies in this dataset make the temporal scale
of sustained NPQ accumulation inconsistent. A less straightforward photophysical phenomena is how the maximum and
minimum dark-adapted fluorescence yields individually respond to temperature, which would describe a separation of
sustained NPQ and photoinhibited reaction centers (i.e., qL is less than 1 even when leaves are fully dark adapted). This
separation is also connected to state transitions between PSII and PSI (Baker et al., 2007; Rath et al., 2022). Biophysiochemical
effects are tied to changes that may be informed by WUE strategies or damage to leaf cellular integrity (Kadir et al., 2006), as
well as membrane and enzyme degradation (Schrader et al., 2004). In principle there would be both a hot and cold temperature
beyond which intra-cellular mechanisms break down, bringing $\Phi_{PSIImax}$ to zero, though they likely vary with life strategy and
phenology (i.e., evergreen needleleaf in cold temperatures (Corcuera et al., 2011)). In future work, findings along this line
would provide physically justified bounds on allowed parameter values, preventing features like the cold temperature result of
BET-Te (Fig. 3d).

The PFT-informed parameterizations of $\Phi_{PSIImax}$ responses to temperature were able to demonstrate a trade-off
strategy of temperature tolerance and resilience of $\Phi_{PSIImax}$ values among 12 commonly used PFTs in TEMs and ESMs.
However, the four PFTs of the original 16 that did not have sufficient data to be described were all boreal plants: needleleaf
deciduous boreal tree (NDT-Bo), broadleaf deciduous boreal tree (BDT-Bo), broadleaf deciduous boreal shrub (BDS-Bo), and
C3 Arctic grass (C3-AG). Since the boreal region was expected to experience more extreme temperature and climate events
(Francis & Vavrus, 2015), understanding how boreal ecosystems respond to changing temperatures became more important
and needs to be addressed in the future. Among the 12 parameterized PFTs, C4 crops were underrepresented in terms of data
and also need to be further studied. Moreover, considering the number of diverse crop species and increased engineering
selection of crop genotypes (Leister, 2022), future studies need to better investigate species/genotype-specific temperature
tolerances and resiliencies of photochemical efficiency for main food and commercial crops.

The CTI-informed parameterization of $\Phi_{PSIImax}$ responses to temperature was limited by the distribution of the original
field-site sub-datasets geographic distribution, which were concentrated in the mid-latitudes and had little data in the highly
productive tropical and boreal forest regions. This data limitation resulted in uncertainty in the assessment of the predictive
power of CTI-informed parameterization in the tropical and boreal forest regions. This uncertainty can be addressed in the
future with new datasets to allow for more robustly parameterized CTI-informed temperature-$\Phi_{PSIImax}$ relationships in these
regions.

A potential complication in comparing the PFT-specific and CTI-informed parameterization is a difference in the
temperature regimes under which data were collected. The data used in the PFT-specific results include plants growing in





greenhouse lacking experienced climatological variation, such that this initial 'shock' towards homeostasis may be dealt with using mostly quick-reacting methods such as energy-dependent NPQ, as no previous temperature acclimation had been required. As plants acclimate to a specific climatology, they shift from reversible to irreversible photoinhibitory strategies, primarily sustained NPQ (Rizza et al., 2001, Rapacz et al., 2004, Ehlert & Hincha, 2008). Future comparison between the two

parameterizations calls for a more consistent and extensive dataset covered by diverse plant species/PFTs under diverse natural habitats.

Broader application of this methodology to other energy partitioning pathways of light reactions is required to fully link light use efficiency with GPP (Gu et al., 2019). $\Phi_{PSIImax}$ (Fv/Fm) is a ratio composed of the minimum and maximum level of chlorophyll fluorescence from a dark-adapted leaf; each is composed of slightly different rate constants of the energy

quenching pathways (Tietz et al., 2017). Further isolation of the temperature-dependent changes between these two variables may allow clarification if the decline in $\Phi_{PSIImax}$ is due to a rise in energy-independent NPQ or a change in the availability of PSII reaction centers for photochemistry. There was an indication within the dataset that the length of temperature exposure also affects temperature-$\Phi_{PSIImax}$ response, which requires further examination. The connected dynamics of water and heat stress have been examined (Ogaya et al., 2011; Ashraf & Harris, 2013; Seng et al., 2023; Sommer et al., 2023), but extensive

PFT-specific relationships and climatological impacts need to be explored in the future.

## 5 Conclusion

- The decline of $\Phi_{PSIImax}$ outside of an ideal temperature range is a consistent response across 146 species covering diverse climatological conditions. We introduce a model to describe this temperature response with easily interpretable parameters.

- There was variability in both the range of temperatures under which PFTs maintained a maximum $\Phi_{PSIImax}$ (tolerance) and the rate of decline outside of the range (resilience). More temperature-resilient PFTs were less temperature tolerant, and vice versa.

- Temperature responses along the tolerance-resilience trade-off suggest three categories of life history strategies of light partitioning to hot and cold extremes: the cold and hot temperature tolerant PFTs (BES and C3-NAG), the cold

and hot temperature resilient PFTs (BDT-Te, NET-Bo, C4-G, C3-C), and the temperature specialists PFT group (BET-Te, BDT-Tr, BET-Tr, C4-C, and BDS-Te).

- Indices of climatological temperature (CTI) variability in space were found to explain some of the variations not captured in the PFT-specific parameterizations of the temperature-$\Phi_{PSIImax}$ relationship alone. We leveraged this into a climatology-informed CTI scheme that was able to improve predictive power in certain regions compared to the

PFT-specific schemes.

- The global distribution of CTI suggests that the regions in which the CTI-informed parameterization performs better compared to the PFT-specific parameterizations fall in the transition zones from temperate to tropical regions and



from the mid-latitudes to the polar region. This is likely tied to experienced more variable climatological environment in the regions promoting a more consistent community temperature response via acclimation.

▪    The advantage of CTI-informed parameterization over PFT-specific parameterizations is PFT-dependent and varies across latitudes. Climatological regulation on the temperature feedback of $\Phi_{PSIImax}$ is critical for PFTs with broad distributions in diverse habitats or those living in local regions with a large seasonal variability in temperature.

**Appendix A | Monte Carlo scheme for determining parameter constraints in the fitting function.**

Differences in available data within each PFT-specific sub-dataset may induce uncertainty in estimated parameters in Eq. 1

and make the fitted parameters for each PFT-specific temperature response function of $\Phi_{PSIImax}$ value incomparable. To avoid this parameterization uncertainty, we imposed unified constraints on each parameters' range using a Monte Carlo scheme. We performed three tests. First, we did a test that involved dividing all paired $\Phi_{PSIImax}$ and temperature data within the dataset of each PFT into 8 subsets. Each subset covers a 10-degree range of measured temperature beginning at -17°C. Each subset then underwent a permutation, with a percentage of the subsets data selected randomly. Each random group of $\Phi_{PSIImax}$ and

temperature measurements in a PFT-specific subset was aggregated and fitted to the model (Eq. 1), and the parameters were recorded. The permutation, random selection, and function fitting were all repeated for 700 iterations. This Monte Carlo scheme was done with three different percentages (75%, 50%, 33%) of data taken from the PFT-specific subsets, to observe the sensitivity of the fitting results to the amount of fitted data. The estimated parameter values from 2100 instances of fitting were finally integrated to describe the distribution range of each parameter. The parameter constraints within the range between

the mean of each parameter ± 2σ, here σ refers to the standard deviation of each parameter's distribution, were recorded. This gave a view of what a parameterization with wide temperature ranges may produce but could be heavily biased by the most extreme subsets with limited data.

For the second test, a similar process was performed, except there were three subsets capturing the relatively constant central temperature range [7-35°C] and the outer decline in cold and hot temperatures. 10% of each subdivision's data was selected

to produce aggregated data for fitting Eq. 1 containing the same number of data points as the average PFT-specific sub-dataset. 2000 iterations of aggregated data points were modeled to produce ranges of parameters that had less chance of being biased by the most extreme data points. However, there was a spreading of modeled values to the point of being unphysical in their interpretation, likely due to aggregated data points that did not actually have cohesive temperature-$\Phi_{PSIImax}$ dynamics.

The third test analysed the distribution of temperature-$\Phi_{PSIImax}$ data pairs with $\Phi_{PSIImax}$ around $a/2$. The spread of temperature

values about this boundary serves as the bounds implicit in the dataset itself and proved to be more physically realistic. By comparing the minimum and maximum temperature values of the $a/2$ $\Phi_{PSIImax}$ data points in cold and hot temperature extremes to the parameter distributions in the previous tests, the parameter space in Table A1 was chosen.

| Parameter | Constrained Range |
|---|---|





| | |
|---|---|
| $a$ | 0.74 - 0.83 |
| $m_1$ | -23 - 7 |
| $s_1$ | 1 - 25 |
| $m_2$ | 35 - 57 |
| $s_2$ | 1 - 13 |

Table A1 Constrained range of fitted parameters using the Monte Carlo scheme

**Appendix B | Methodology for performing the Aligned Rank Transform**

We performed ANOVA to evaluate how climatology metrics affect the prediction efficiency of the developed PFT-specific temperature-$\Phi_{PSIImax}$ function. Before assigning ranks and performing the standard ANOVA analysis, we performed several preprocessing steps that isolated each interaction term between SMET, WMET, and AAT. First, the aligned X values for the main effect of SMET, WMET, and AAT were found using Eqs. B1-3.

$$\text{SMET}' = (X - Y_{ijk}) + (\overline{\text{SMET}_i} - \mu) \qquad \text{Eq. B1}$$

$$\text{WMET}' = (X - Y_{ijk}) + (\overline{\text{WMET}_j} - \mu) \qquad \text{Eq. B2}$$

$$\text{AAT}' = (X - Y_{ijk}) + (\overline{\text{AAT}_k} - \mu) \qquad \text{Eq. B3}$$

Here $Y_{ijk}$ is the 'cell mean', calculated as the mean of all X values in the test as the $i^{th}$ level of SMET, the $j^{th}$ level of WMET, and the $k^{th}$ level of AAT as found using Eq. 6. The $\mu$ is the mean of all X values used in the test. $\overline{\text{SMET}_i}$, $\overline{\text{WMET}_j}$, and $\overline{\text{AAT}_k}$

are the mean value of X at the given level of SMET, WMET, and AAT respectively.

The aligned X values for the two-way effects of SMET, WMET, and AAT were found using Eqs. B4-6.

$$\text{SMET}' \times \text{WMET}' = (X - Y_{ijk}) + (\overline{\text{SMET}_i\text{WMET}_j} - \overline{\text{SMET}_i} - \overline{\text{WMET}_j} + \mu) \qquad \text{Eq. B4}$$

$$\text{SMET}' \times \text{AAT}' = (X - Y_{ijk}) + (\overline{\text{SMET}_i\text{AAT}_k} - \overline{\text{SMET}_i} - \overline{\text{AAT}_k} + \mu) \qquad \text{Eq. B5}$$

$$\text{WMET}' \times \text{AAT}' = (X - Y_{ijk}) + (\overline{\text{WMET}_j\text{AAT}_k} - \overline{\text{WMET}_j} - \overline{\text{AAT}_k} + \mu) \qquad \text{Eq. B6}$$

The aligned X values for the three-way interaction were found using Eq. B7.

$$\text{SMET}' \times \text{WMET}' \times \text{AAT}' = (X - Y_{ijk}) + (\overline{\text{SMET}_i\text{WMET}_j\text{AAT}_k} -$$
$$\overline{\text{SMET}_i\text{WMET}_j} - \overline{\text{SMET}_i\text{AAT}_k} - \overline{\text{WMET}_j\text{AAT}_k} + \overline{\text{SMET}_i} + \overline{\text{WMET}_j} + \overline{\text{AAT}_k} - \mu) \qquad \text{Eq. B7}$$

$\overline{\text{SMET}_i\text{WMET}_j}$ is the mean value of X at $i^{th}$ level of SMET and $j^{th}$ level of WMET. $\overline{\text{SMET}_i\text{AAT}_k}$ is the mean value of X at $i^{th}$ level of SMET and $k^{th}$ level of AAT, and $\overline{\text{WMET}_j\text{AAT}_k}$ denotes the mean value of X at the $j^{th}$ level of WMET and $k^{th}$ level of





SMET. The term $\overline{\mathrm{SMET}_i\,\mathrm{WMET}_j\,\mathrm{AAT}_k}$ is equivalent to the 'cell mean'. These terms were then used to perform the standard ANOVA analysis.

**Appendix C | Algorithm for smoothing fitted parameters and corresponding central CTI values of QSA sets**

To decrease the existence of noisy data impact on the regression of fitted parameters on the central CTI values of QSA sets, we smoothed the fitted parameters and corresponding central CTI values of the QSA sets before performing the final regression

analysis. To determine a smoothing window width, we defined the 'distance' between two QSA sets as the difference between their central CTI values, then integrated all distances between all combinations of QSA sets to determine their histogram distribution (Figure C1). The median CTI value of this distribution, hereafter referred to as the Median Intra-point Distance (MID), was 367 and then used as the smoothing window width about a central CTI value q. Each individual QSA set-fitted parameter ($m_1$, $m_2$, $s_1$, $s_2$) from Eq. 1 was then adjusted to the mean of the corresponding parameter estimates within q ± MID.


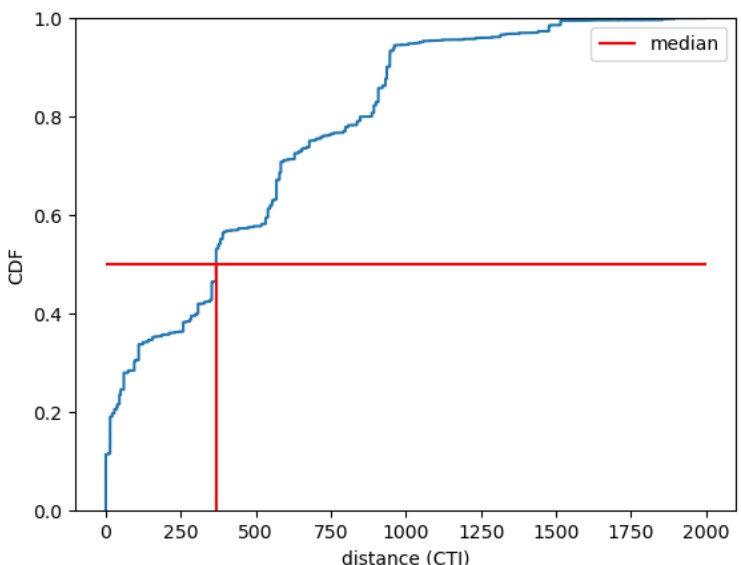

Figure C1. CDF of the distances between central CTI values of the QSA sets.

**Code availability**

Codes for reproducing the parameter results for the PFT-specific and CTI-informed schemes and results visualization are

publicly available at the University of Arizona (UA) research data repository ReDATA (DOI: 10.25422/azu.data.24143064), after the manuscript is published.





**Data availability**

This work was based on collecting and synthesizing data from published peer-reviewed references. The synthesized global-scale maximum photochemical quenching yield and corresponding environmental dataset is publicly available at the University

of Arizona (UA) research data repository ReDATA (DOI: 10.25422/azu.data.24142989). The CRUNCEP v.7 data used for climate data is publicly available (DOI: https://rda.ucar.edu/datasets/ds314.3/; Viovy et al, 2018). Plant functional type distribution data (Lawrence et al., 2016) is available at the UA research data repository ReDATA (DOI: 10.25422/azu.data.24143064).

**Author contribution**

YS designed the research and all authors contributed to the conceptualization of the study. PN collected, quality controlled, and synthesized the published PAM fluorometry dataset. YS and PN developed the modelling pipeline, and PN performed the model work. Interpretation of results was developed through discussion between all authors. PN wrote the initial manuscript with primary revisions and comments from YS and GL.

**Competing interests**

The contact author has declared that none of the authors has any competing interests.

**Acknowledgments**

YS is supported by the start-up funding of the University of Arizona (UA). LG are supported by the U.S. Department of Energy
(DOE), Office of Science, Biological and Environmental Research (BER) Program. ORNL is managed by UT-Battelle, LLC, for DOE under contract DE-AC05-00OR22725. PN is mainly supported by YS's UA start-up funding and partially supported by US DOE BER program.

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
