# Peer review of "The Effect of Temperature on Photosystem II Efficiency across Plant Functional Types and Climate"

_Biogeosciences, 2023_

## Author Comment (AC2)

**Response to Reviewer Comments for DOI: 10.5194/bg-2023-163**

___Anonymous Referee #2___*: Neri et al synthesized PAM measurements of ΦPSIImax from literature, and investigated its temperature responses. A model with interpretable parameters is developed, and a tolerance-resilience trade-off is identified. The impacts of PFT and climatological temperature on ΦPSIImax tolerance and resilience are also investigated. While plenty ΦPSIImax measurements can be found in literature, a synthesis analysis as this work is absent. The presented work could be valuable to the community by facilitating our understanding of photosynthesis temperature response and providing information for model parameterizations. The manuscript effectively presented the methods and results in general. Below lists my several concerns and suggestions.*

**Response:** We thank this referee for appreciating our work and for such a thorough reading of the manuscript. The comments and suggestions made in this review are very helpful in guiding us to improve the text of our manuscript. We have responded to each point as detailed below.

1. *It is not clear to me how the confounding variables (water, light, etc.) were controlled, although that is stated as a selection criterion (L125). Did you select studies where the confounding variables were controlled in that specific study? My understanding is those variables can still vary from one study to another, and may play a role in the analyses. Could you clarify this?*

**Responses:** We appreciate the reviewer's question about the selection criteria. We selected data with the control of "no other stresses" (e.g., water, light, nutrients). These details are described in the dataset collection document (cited on L130), but we neglected to include the relevant information in the main text. In the appendix dataset, we use "light_status=0", "water_status=0", and "nut_status=0" to label data measured under no light, water, and nutrient stresses respectively. For light status, an additional consideration with some publications was given if the measurement was taken in a climate exposed condition, in which case it may be given a "light_status=1" but still included in the used data for modeling. Among all 2330 measurement data points from the selected 104 studies, 2204 measurement data points met these criteria (Figure 1c), i.e. from PAM monitoring in controlled environments (e.g., greenhouse with no stress of light, water, and nutrients) or field experiments with the description of no other stress condition. To clarify, we will revise the description on lines 123-126, as well as line 150.

**On lines 123-126:** "To isolate temperature dependence from other external regulators of $\Phi_{PSIImax}$, we mined and selected data from studies that provided cohesive descriptions of temperature for the relevant measurements and excluded the effects of other confounding variables (e.g., water, nutrient, light stress). Following this data selection strategy, we selected PAM observations from the controlled environments (e.g., green house) where nutrients, lights, and water availability have been optimized and only varied temperatures are considered. We also included PAM data from field experiments with the description of no other stress conditions except for temperature. Following these guidelines, a total of 104 studies out of the 380 publications were finally selected."

**On line 150:** "In total, 2330 measurements from 104 studies were recorded in the final database, with 2104 measurements meeting the criteria for use in modeling."

2. *PFT-specific CTI and percent prediction explained (Eq. 7 and Figure 5): My understanding is that the PFT-specific CTI is still one equation generated for all PFTs, rather than one equation for each PFT. Is this correct? Could you explain the reason for using a general*

*equation instead of one equation for each PFT? Presenting the values of the aL parameters might also be helpful.*

**Responses:** This question concerns the description of the three terms about the "CTI" informed parameterization, including "PFT-specific CTI" and "general CTI" shown in Figure 5, as well as "CTI-informed temperature-$\Phi_{PSIImax}$ function" as shown in Figure 6. "PFT-specific CTI" in Figure 5 (green bars) presented the values of estimated aL parameters in Eq. 7. We should have given a clearer legend for this information (e.g., "ART ANOVA results with PFT-specific model residuals"). These values are calculated by performing one ART ANOVA analysis, which determined the residuals (X) between the collected $\Phi_{PSIImax}$ values ($\Phi_{PSIImax,O}$) and predicted $\Phi_{PSIImax}$ values ($\Phi_{PSIImax,P}$) given by each PFT-specific temperature-$\Phi_{PSIImax}$ function. These residuals are integrated to estimate the contributions from different temperature metrics individually as well as from the interactions among them to the overall prediction errors of the 12 PFT-specific temperature-$\Phi_{PSIImax}$ functions (Lines 250-251).

In contrast, "general CTI" in Figure 5 (blue bars) refers to aL values from the second ART ANOVA analysis, which examines the contribution of individual temperature metrics and the interactions among them to the prediction residuals by the general temperature-$\Phi_{PSIImax}$ function derived using all data in the field site sub-dataset (Lines 251-254). Therefore, a clear legend of this second ART ANOVA in Figure 5 should be "ART ANOVA using prediction residuals from general (non-PFT specific) temperature-$\Phi_{PSIImax}$ function."

Comparing the estimated aL values from the two ANOVA analyses in Figure 5 aims to examine whether the contributions of individual temperature metrics and their interactions to prediction errors are consistent, and to provide justification and flexibility for applying either version of aL values for CTI estimations (Lines 459-463).

As the two versions of ANOVA analysis showed consistency, we applied aL values from the PFT-specific prediction residuals-based ART ANOVA (green bars) to estimate CTI values corresponding to each $\Phi_{PSIImax}$ value in the field site sub-dataset, using Eq. 7. Then, these CTI values are incorporated to quantify the dependence of the parameters from PFT-specific temperature-$\Phi_{PSIImax}$ functions ($m_1$, $m_2$, $s_1$, $s_2$) on CTI, using quantile system approach (QSA) (Section 2.3.4). For each parameter ($m_1$, $m_2$, $s_1$, $s_2$), we generated one CTI-informed function using all data from the field site sub-dataset. The reason for generating one equation for each parameter using all data from the field site sub-dataset is as follows. (1) A lack of sufficient data covering a large range of CTI values for any one PFT in the field site sub-dataset meant that CTI-dependence of each parameter ($m_1$, $m_2$, $s_1$, $s_2$) for each PFT could not be determined in a statistically robust way (Lines 285-287). (2) Moreover, the use of a single equation and comparison to the PFT-specific temperature-$\Phi_{PSIImax}$ functions can test the core hypothesis in this study: "climatological temperature regulates the temperature tolerance and resilience of $\Phi_{PSIImax}$, therefore shifts different PFT's temperature-$\Phi_{PSIImax}$ responses toward converged responses to the climatology of their "similar" local habitat. (Lines 209)"

In summary, we accept the reviewer's comments and will clarify this concern by revising the following contents of the manuscript.

    1)   We will refine the description of the hypothesis and its testing methods.

    **On lines 209-215:** "To test the hypothesis that climatological temperature regulates the temperature tolerance and resilience of $\Phi_{PSIImax}$, and therefore shifts different PFT's temperature-$\Phi_{PSIImax}$ responses toward converged responses to the climatology of their "similar" local habitat, we generated a general

climatology-informed temperature-$\Phi_{PSIImax}$ function and compared its results with the corresponding PFT-specific model results.

In detail, we quantified corresponding climatological temperature metrics for data within the field site sub-dataset (Sect. 2.3.1) and assessed their capacity to explain the prediction residuals from PFT-specific temperature-$\Phi_{PSIImax}$ functions using ART ANOVA (Sect. 2.3.2). Based on the results, we incorporated the metrics via a linear combination into a Climatology Temperature Index (CTI) (Sect. 2.3.3). This index was then incorporated to quantify a CTI-informed temperature-$\Phi_{PSIImax}$ function (Sect. 2.3.4). The fitting results of this CTI-informed model were compared to the corresponding PFT-specific model results. Finally, we identified where prediction deficiency was improved by the CTI-informed parameterization and the climatology's effect on the temperature-$\Phi_{PSIImax}$ relationship was important to consider (Sect. 2.3.5)."

2) We will refine the description of the reasons for generating one equation for each parameter using all data from the field site sub-dataset after lines 285-286.

**On lines 285-286:** "Ideally, the field site sub-dataset would cover diverse climatological temperature conditions, be distributed consistently across the full global range of CTI values, and contain statistically sufficient data for all PFTs, but this is not the case. The available 709 measurements represent a limited, non-uniform range of climatology temperature metrics (Histogram distribution of data in Fig. 2b). We overcome this data limitation by generating one CTI dependence function for each parameter in Eq. 1 using all data from the field site sub-dataset and the quantile system approach (QSA), which was developed to navigate the small sample size and inconsistent CTI values distribution by performing the following three steps."

3) To avoid confusion about the two versions of ART ANOVA and clarify the specific ART ANOVA finally employed to generate CTI-informed parameterization, we will revise their description in Figure 5 and the corresponding texts in section 3.2.1.

"The green bars" in Figure 5 are defined as "ART ANOVA using prediction residuals from PFT-specific temperature-$\Phi_{PSIImax}$ functions (ANOVA$_{RS\_pft}$)", whereas "the blue bars" in figure 5 are labeled as "ART ANOVA using prediction residuals from the general (non-PFT specific) temperature-$\Phi_{PSIImax}$ function (ANOVA$_{RS\_gen}$)".

**On lines 461-462:** "This consistency justified that the regulation of climatological temperature on the temperature-$\Phi_{PSIImax}$ relationship can be estimated using the results of either version of ANOVA. Here we will use the aL values from ANOVA$_{RS\_pft}$."

3. *Rearranging Section 3.2 and Section 3.3 and putting the CTI map (Figure 8c) before Figures 6 and 7 may help the audience more easily interpret results related to CTI.*

**Responses:** We agree with the reviewer that this rearranging of sections and figures can help to interpret the main results related to CTI and benefit our readers in capturing key points of this study. To address this, we will perform a rearrangement of section 3.2 & section 3.3 and corresponding figures, with more details to be found in the response to **(5)** below.

4. *Are the CTIs in the results section the general CTI?*

**Responses:** No, all CTI values after 3.2.1 were generated using the aL weights estimated from ART ANOVA using prediction residuals from the PFT-specific functions. As discussed in our responses to **Comment #2**, this is a point that we should have made clearer, and we will add language to address this around L462.

**On Lines 461-462:** "This consistency justified that the regulation of climatological temperature on the temperature-$\Phi_{PSIImax}$ relationship can be estimated using the results of either version of ANOVA. Here we will use the aL values from ANOVA$_{RS\_pft}$."

5. *The manuscript is quite long, I suggest cutting the length of the manuscript. Some method and results could potentially be moved to the supplementary. For example, details of ART ANOVA, section 3.2.1, and section 3.2.3.*

**Responses:** This comment helps us polish our manuscript. We agree that with some re-wording and summary of the main points, section 3.2.1 and section 3.2.3 could be joined into Appendix B. In the meantime, we will combine sections 3.2 and 3.3 and re-organize the text, including three sub-sections:

3.3.1 Global distribution of CTI: This section will focus on the description of the current section 3.3.1 (CTI global pattern) and also include a brief justification of CTI for explaining the prediction errors (a brief summary of the main conclusion from current section 3.2.1).

3.3.2 CTI-informed parameters of temperature regulation on $\Phi_{PSIImax}$ and its latitudinal variation: This section will combine the current section 3.2.2 with section 3.3.2.

3.3.3 Spatial distribution of the differences between CTI-informed and PFT-specific parameterizations: This section will briefly report the "overall improvement of predictions of the temperature-$\Phi_{PSIImax}$ dynamic using CTI-informed parameterization (Figure 7 in current section 3.2.3) and focus on description of Spatial distribution of the differences between CTI-informed and PFT-specific parameterizations (current section 3.3.3).

---

## Author Response (AR1)

**Responses to the Editor and Reviewers' Comments**
*Editor's comments*
*Thank you for conducting an interactive discussion with two anonymous referees. Both referees evaluated that this study has good to excellent scientific significance and good presentation quality. Based on the discussion, the scientific quality of the manuscript would be improved, as found in your detailed responses to the referee's comments. For example, distinguishing response and acclimation would improve clarity, and refining the description of the hypothesis would improve scientific quality. This study would potentially attract attention from a wide range of vegetation model researchers by providing a parameterization of photosynthetic acclimation to temperature, e.g., climatic warming. As both referees suggested, the manuscript will be reconsidered after revisions.*

**Response:** Thank you for providing this opportunity to improve our manuscript. Following two referees' valuable and helpful comments, we have revised the manuscript by implementing our previous responses to the reviewers' comments. Please see our detailed responses to reviewers' comments, including a list of revised contents corresponding to each comment. ***Please note that the line numbers are slightly different in the revision-tracked version and the non-tracked version. Here, we provided two types of line numbers for your convenience. Take "Lines 1-3 (Tracked version Lines 4-5)" as an example; the former refers to the line number in the untracked manuscript (the formatted version), and the latter refers to the line number in the tracked manuscript.***

*Anonymous Referee #1:* *The manuscript by Neri et al. explored how maximum PSII yield changes with PFT and climate using data collected from the literature. The research topic was of great importance for the global carbon cycle, and implementing the idea in terrestrial biosphere models will help improve the model predictions. The manuscript was overall well written, and ideas were well-delivered. While I am convinced about the importance of the idea, I have some concerns about the research and analyses performed. Below are two primary issues I found, and I hope they are useful for the authors.*

**Response:** We appreciate this reviewer's comments and careful reading of the manuscript, and the insights provided to us. We have carefully considered each question raised and will revise the manuscript accordingly. Our responses to each specific comment are as follows.

1.    *Simply modifying Phi_PSIImax is not adequate for photosynthesis and thus fluorescence models. For example, if the change of Phi_PSIImax is due to those of the rate constants, such as Kd, Kf, Kn, and Kpmax, prescribing Phi_PSIImax will only impact the calculation of electron transport rate J and thus Aj and Agross. However, the subsequent qL, NPQ, and Phi_f calculations will not be accurate as the Kd/f/n/pmax are not changing accordingly. Therefore, a more process-focused model to explain Phi_PSIImax will be more useful. For example, the van der Tol et al. (2013) fluorescence model assumed that Kd is temperature-dependent to explain the temperature dependency of Phi_f on temperature. A similar approach, such as a revised Kn (temperature) function, can be taken here.*

**Response:** We agree with the reviewer that prescribing $\Phi_{PSIImax}$ as a function of temperature only directly impacts the estimation of $A_J$, and a more process-focused model, such as the van der Tol et al. (2013) or the Gu et al. (2023) approach, will be more useful for parameterizing overall effects of temperature on photosynthesis and thus fluorescence models. Following the van der Tol et al. (2013) approach, integrating our global-scale PAM datasets to parameterize the temperature-Kd/f/n/pmax function is straightforward. For example, the temperature function of $\Phi_{PSIImax}$ developed in our study can be directly applied into the $x$ function in van der Tol et al. (2013), which would enable the simulation of Kn variation with temperature.

However, as already alluded by this reviewer, photosynthesis is a multi-stage phenomenon. Gu et al. (2023) separated photosynthesis into three stages of reactions – photophysical reactions, photochemical reactions, and biophysical reactions. These three stages have both mutually dependent and independent reactions, including temperature responses. This means that a temperature response in one stage can appear as if it is in a response in another stage. The ability to distinguish independent from dependent temperature responses is essential in this approach and will have to be achieved through mechanistic process understanding. Enzymatic reactions of biochemistry, e.g., carboxylation and oxygenation, have both well-understood and well-quantified temperature responses (*i.e.*, the Farquhar biochemical model of photosynthesis). The Marcus theory of electron transfer in proteins can be used to similarly describe the temperature response of the photochemical reactions (Gu et al. 2023). Currently, however, the precise mechanisms of temperature response of photophysical reactions, which include those of different rate constants that directly affect $\Phi_{PSIImax}$ and thus are important to the present study, are not well understood.

We believe that an empirical parameterization of the independent temperature response of $\Phi_{PSIImax}$ is an effective strategy for modeling the temperature effects of photophysical reactions because $\Phi_{PSIImax}$, which is equal to Kpmax/(Kf + Kd + Kni + Kpmax), is an integrative quantity of photophysical reactions and is key to modeling J, $A_J$, and Agross. Here Kf, Kd, Kni, and Kpmax are the rate constants for fluorescence, constitutive heat dissipation, energy-independent non-photochemical quenching (NPQ), and photochemistry when PSII reaction centers are fully open. Without an understanding of the processes that may control the dynamics of these rate constants, we are concerned that empirically parameterizing temperature responses of individual rate constants may run into the risk of mixing the independent and dependent temperature responses and lead to erroneous interpretations. This is a legitimate concern because none of the rate constants can be monitored directly under natural conditions. $\Phi_{PSIImax}$ can be monitored directly, however. A conservative strategy at present is to treat Kf and Kd as physical properties of pigment molecules and assume they are insensitive to temperature under typical physiological conditions (see for example curve a in Fig 1 of Pospisil et al. 1998; Fig 6 of Tesa et al. 2018). Note that even if Kf or Kd have no temperature dependence, the energy allocated to fluorescence or constitutive heat dissipation *in vivo* can still be temperature dependent because of the coupling between different energy dissipation pathways and because of the feedbacks from the photochemical and biochemical reactions on the photophysical reactions.

A full modeling of temperature responses of photosynthetic variables, including qL, NPQ, and $\Phi_F$, can be achieved by coupling the photophysical reactions (Gu et al., 2019), photochemical reactions (Gu et al., 2023), and the Farquhar biochemical model (Farquhar et al., 1980), with the support of the temperature dependence modeling of $\Phi_{PSIImax}$ provided by this study. We are currently still working on this coupling, which is a large undertaking, and is beyond the scope of this current study. **The present specific study aims** to (1) provide a global scale

parameterization of temperature responses of $\Phi_{PSIImax}$ and its variability across plant functional types and illuminate a so-far poorly understood dynamic trade-off between tolerance and resilience of the temperature-$\Phi_{PSIImax}$ relationship and (2) demonstrate how incorporating climatology into analysis of the temperature-$\Phi_{PSIImax}$ relationship can improve the prediction of $\Phi_{PSIImax}$. Acquiring this knowledge is important for understanding and predicting temperature regulation on electron transport rates and $A_J$, which has been underrepresented in the current photosynthesis model (e.g., Farquhar model, Collatz model), compared with the thorough consideration of temperature controlling photosynthetic capacity parameters (Vcmax) and biochemical kinetics parameters. Moreover, understanding the differences in tolerance and resilience of the temperature-$\Phi_{PSIImax}$ relationship among different PFTs will facilitate our assessment of the photosystem II efficiency of diverse PFTs under climate change and climate extremes.

Considering this manuscript is already fairly long, we prefer not to include integrated parameterization of the temperature-Kd/f/n/pmax function. Instead, we will integrate the reviewer's suggestion with our ongoing effort, which is employing the global-scale PAM dataset and temperature-$\Phi_{PSIImax}$ functions from this study to parameterize the integrated temperature effects on light partitioning and photosynthesis in the fluorescence-enabled photosynthesis model, as described in our previous study (Gu et al., 2019). To respond to the reviewer's comment, we will revise the introduction **(Lines 100-119, Tracked version lines 100-119)** to better refine the scope and aim of this study. We have also revised the discussion session 4.5 Uncertainty and future work **(Lines 752-757, tracked version lines 841-846)** to highlight that this is only the first step on the road of mechanistic fluorescence-enabled photosynthesis modeling. We will outline the subsequent work built upon this study.

Farquhar, G.D., von Caemmerer, S. & Berry, J.A. (1980) "A biochemical model of photosynthetic CO2 assimilation in leaves of C3 species." Planta **149**: 78-90.

Gu, L., et al. (2019). "Sun-induced Chl fluorescence and its importance for biophysical modeling of photosynthesis based on light reactions." New Phytologist **223**(3): 1179-1191.

Gu L, Grodzinski B, Han J, Marie T, Zhang Y-J, Song YC, Sun Y. 2023. "An exploratory steady-state redox model of photosynthetic linear electron transport for use in complete modeling of photosynthesis for broad applications." Plant, Cell and Environment **46**: 1540-1561.

Pospíšil P, Skotnica J, Nauš J (1998) Low and high temperature dependence of minimum F0 and maximum FM chlorophyll fluorescence in vivo. Biochimica et Biophysica Acta 1363 1998 95–99.

Tesa M, Thomson S, Gakamsky A (2018) Temperature-dependent quantum yield of fluorescence from plant leaves. AN_P41, Edinburgh Instruments.

According to our responses, the revised texts in the manuscript include:
**On lines 100-119 (Tracked version lines 100-119):** "Our previous effort (Gu et al., 2019) has modelled the leaf-level SIF-GPP dynamics as a function of NPQ, qL, $\Phi_{PSIImax}$, and absorbed photosynthetically active radiation (APAR). That study pointed out a need for mechanistic

descriptions of how NPQ, qL, and $\Phi_{PSIImax}$ respond to environmental conditions to accurately predict environmental regulation of the GPP-SIF relationship at the leaf level. By empirically fitting the NPQ rate coefficient with a function of relative light saturation and combining it with the biochemical reactions-centred photosynthesis model, van der Tol (2014) estimated the responses of leaf-level fluorescence yield to changing temperature, light, and $CO_2$ concentration, indicating that quantifying environmental responses of photochemical yield are a key step in addressing the integrated environmental impacts on SIF-GPP dynamics. Therefore, here we present a novel model of $\Phi_{PSIImax}$ response to temperature variation by collecting and applying a global-scale database of published PAM measurements, with an emphasis on parameterizing the different temperature tolerance and resilience of various plant functional types (PFTs) and investigating how habitat climatology may affect this temperature-$\Phi_{PSIImax}$ relationship. This study will deliver the first global-scale quantification of temperature impact on photosystem II efficiency and its variability across PFT and habitat climatology and build a theoretical basis for assessing vegetation light utilization potential for carbon sequestration under climate change and climate extremes. Modelling temperature regulation on $\Phi_{PSIImax}$ is important for assessing extreme temperature impacts on the maximum electron transport rate (Jmax) in biochemical reactions-centred photosynthesis models. Moreover, characterizing the temperature response of $\Phi PSIImax$ will allow us to connect other light partitioning mechanisms to temperature change, building the first step of resolving coupled SIF and GPP responses to temperature change. With the support of the temperature dependence modelling of $\Phi_{PSIImax}$ provided by this study, a full modelling of temperature responses of photosynthetic variables, including qL, NPQ, and $\Phi F$, can be achieved by coupling the photophysical reactions (Gu et al., 2019), photochemical reactions (Gu et al., 2023), and the Farquhar biochemical model (Farquhar et al., 1980).

**On lines 752-757 (Tracked version lines 841-846):** "$\Phi_{PSIImax}$ (Fv/Fm) is a ratio composed of the minimum and maximum levels of chlorophyll fluorescence from a dark-adapted leaf (Tietz et al., 2017). In future work, we will further isolate the temperature-dependent changes between these two variables and link the derived temperature-$\Phi_{PSIImax}$ functions in this study with the estimation of relative light saturation and rates of other energy dissipation pathways. These future efforts will allow clarification if the decline in $\Phi_{PSIImax}$ is due to a rise in energy-independent NPQ or a change in the availability of PSII reaction centers for photochemistry."

2.  *The authors did not distinguish "response" and "acclimation" in the analyses. For example, let us again assume Phi_PSIImax change is due to those of Kd, Kf, Kn, and Kpmax here. If Kd = a1\*T + b1 for plants grown in the C1 environment and Kd = a2\*T + b2 for plants grown in the C2 environment, the function a\*T + b is "response" (related to temporary changes in the environment), and shift from a1\*x + b1 to a2\*x + b2 is "acclimation" (related to long term changes in climate). Therefore, it is likely that the data analyzed is a mixture of "response" and "acclimation", and attributing all the changes in Phi_PSIImax is inappropriate. Without distinguishing the two, the analyses performed might be biased.*

**Responses:** Thanks for a clear explanation of "response" vs "acclimation". This is a great point. Distinguishing response and acclimation is important for clarifying two key results from this manuscript. First, the temperature-$\Phi_{PSIImax}$ function developed for each PFT is referred to as the "temperature responses" of a specific PFT (Section 3.1). Second, Section 3.2 (Climatology influence on the temperature-$\Phi_{PSIImax}$ function) addresses how PFT-specific temperature-$\Phi_{PSIImax}$

responses can "shift" with habitat climatology by quantifying the regression of temperature resilience and tolerance parameters on the climatological temperature index (CTI) (Figure 6) and comparing the differences between these CTI-informed and PFT-specific temperature tolerance metrics and resilience parameters of plant $\Phi_{PSIImax}$ values (Figure 8 in the revised manuscript). The discussion of section 3.2 aims to test a core hypothesis that climatological temperature regulates the temperature tolerance and resilience of $\Phi_{PSIImax}$ 'in the wild', therefore shifting different PFT's temperature-$\Phi_{PSIImax}$ responses toward converged responses to the climatology of their "similar" local habitat. Considering the collected dataset itself does not clearly address if "this shift" may be related to either plant's acclimation or adaptation (evolutionary shift) to habitat climatology, we only describe "this shift" as a potential result of plant acclimation and adaptation to habitat climatology (e.g., on lines 664 **(tracked version: 753)**, 674 **(tracked version: 763)**, 707 **(Tracked version: 796)**, 746 **(Tracked version: 835)**, 779 **(Tracked version: 872)**).

By definition, $\Phi_{PSIImax}$ is supposed to be measured on dark-adapted leaves for which energy-dependent NPQ is zero and all available PSII reaction centers are fully open. This means that the effects of short-term temperature variation are removed by the measurement protocols. In our data gathering from the published literature, we ensured that the following quality measures were met in the studies included in our dataset: an established toolset was used to perform the PAM fluorometry measurements (e.g., a Walz or other industry-standard technology), and a sufficient dark-adaptation time (generally greater than 2 hours) with preference to over-night length dark adaptation of the material before measurement.

However, we realize that the usage of "acclimation" in some locations of the text is not consistent with the above points, including the title, lines 16-18, lines 70-72, lines 120-121, and lines 168-170. We have adjusted these parts of the texts to "response," "impact," or "effect" in a revised version of the manuscript. In addition, we have revised the description of the hypothesis on lines 217-220. A list of our revisions is shown below.

**The title:** "The Effect of Temperature on Photosystem II Efficiency across Plant Functional Types and Climate"

**On lines 16-18 (Tracked version lines 16-18):** "To understand the spatiotemporal variability of $\Phi_{PSIImax}$, we analysed the temperature effect on $\Phi_{PSIImax}$ across plant functional type (PFT) and habitat climatology. The analysis showed that temperature's impact on $\Phi_{PSIImax}$ is shaped more by climate than by PFT for plants with broad latitudinal distributions or in regions with extreme temperature variability."

**On lines 70-72 (Tracked version lines 69-71)**: "However, $\Phi_{PSIImax}$ can be irreversibly downregulated due to plant energy-independent NPQ response to temperature and other environmental stresses, especially extreme temperature, or as a result of photodamage to reaction centers (i.e., qL is less than 1 even when plants are fully dark-adapted (Porcar-Castell, 2011))."

**On lines 120-121 (Tracked version lines 127-128)**: "In this study, we developed specific temperature response functions of $\Phi_{PSIImax}$ for 12 plant functional types (PFTs) commonly used in TBMs and determined temperature 'tolerance' and 'resilience' parameters for $\Phi_{PSIImax}$."

**On lines 168-170 (Tracked version lines 176-783)**: "We employ the PFT-specific sub-datasets to parameterize a general temperature response function of $\Phi_{PSIImax}$ for all data, and 12 PFT-specific temperature response functions. We quantified the temperature tolerance and resilience of $\Phi_{PSIImax}$ for each PFT based on the corresponding parameterized temperature response function."

**On lines 217-220 (Tracked version lines 225-228):** "To test the hypothesis that climatological temperature regulates the temperature tolerance and resilience of $\Phi_{PSIImax}$, and therefore shifts different PFT's temperature-$\Phi_{PSIImax}$ responses toward converged responses to the climatology of their "similar" local habitat, we generated a general climatology-informed temperature-$\Phi_{PSIImax}$ function and compared its results with the corresponding PFT-specific model results."

***Anonymous Referee #2***: *Neri et al synthesized PAM measurements of ΦPSIImax from literature, and investigated its temperature responses. A model with interpretable parameters is developed, and a tolerance-resilience trade-off is identified. The impacts of PFT and climatological temperature on ΦPSIImax tolerance and resilience are also investigated. While plenty ΦPSIImax measurements can be found in literature, a synthesis analysis as this work is absent. The presented work could be valuable to the community by facilitating our understanding of photosynthesis temperature response and providing information for model parameterizations. The manuscript effectively presented the methods and results in general. Below lists my several concerns and suggestions.*

**Response:** We thank this referee for appreciating our work and for such a thorough reading of the manuscript. The comments and suggestions made in this review are very helpful in guiding us in improving the text of our manuscript. We have responded to each point as detailed below.

1. *It is not clear to me how the confounding variables (water, light, etc.) were controlled, although that is stated as a selection criterion (L125). Did you select studies where the confounding variables were controlled in that specific study? My understanding is those variables can still vary from one study to another, and may play a role in the analyses. Could you clarify this?*

**Responses:** We appreciate the reviewer's question about the selection criteria. We selected data with the control of "no other stresses" (e.g., water, light, nutrients). These details are described in the dataset collection document (cited in L138), but we neglected to include the relevant information in the main text. In the appendix dataset, we use "light_status=0", "water_status=0", and "nut_status=0" to label data measured under no light, water, and nutrient stresses, respectively. For light status, an additional consideration with some publications was given if the measurement was taken in a climate-exposed condition, in which case it may be given a "light_status=1" but still included in the used data for modeling. Among all 2329 measurement data points from the selected 104 studies, 2204 measurement data points met these criteria (Figure 1c), i.e. from PAM monitoring in controlled environments (e.g., greenhouse with no stress of light, water, and nutrients) or field experiments with the description of no other stress condition. To clarify, we have added the following descriptions.

**On lines 128-134 (Tracked version lines 135-141):** "To isolate temperature dependence from other external regulators of $\Phi_{PSIImax}$, we mined and selected data from studies that

provided cohesive descriptions of temperature for the relevant measurements and excluded the effects of other confounding variables (e.g., water, nutrient, light stress). Following this data selection strategy, we selected PAM observations from the controlled environments (e.g., green house) where nutrients, lights, and water availability have been optimized and only varied temperatures are considered. We also included PAM data from field experiments with the description of no other stress conditions except for temperature. Following these guidelines, a total of 104 studies out of the 380 publications were finally selected."

**On lines 157-158 (Tracked version lines 165-166):** "In total, 2329 measurements from 104 sites were recorded in the final database, with 2104 measurements meeting the criteria for use in modelling."

2. *PFT-specific CTI and percent prediction explained (Eq. 7 and Figure 5): My understanding is that the PFT-specific CTI is still one equation generated for all PFTs, rather than one equation for each PFT. Is this correct? Could you explain the reason for using a general equation instead of one equation for each PFT? Presenting the values of the aL parameters might also be helpful.*

**Responses:** This question concerns the description of the three terms about the "CTI" informed parameterization, including "PFT-specific CTI" and "general CTI" shown in Figure 5 (now Fig. D1), as well as "CTI-informed temperature-$\Phi_{PSIImax}$ function" as shown in Figure 6. "PFT-specific CTI" in Figure 5 (green bars in Fig. D1 now) presented the values of estimated aL parameters in Eq. 7. We should have given a clearer legend for this information (ANOVA$_{RS\_pft}$ refers to ART ANOVA results with PFT-specific model residuals). These values are calculated by performing one ART ANOVA analysis, which determined the residuals (X) between the collected $\Phi_{PSIImax}$ values ($\Phi_{PSIImax,O}$) and predicted $\Phi_{PSIImax}$ values ($\Phi_{PSIImax,P}$) given by each PFT-specific temperature-$\Phi_{PSIImax}$ function. These residuals are integrated to estimate the contributions from different temperature metrics individually as well as from the interactions among them to the overall prediction errors of the 12 PFT-specific temperature-$\Phi_{PSIImax}$ functions (**Lines 262-263, tracked version lines: 277-278**).

In contrast, "general CTI" in Figure 5 (blue bars in Fig. D1 now) refers to aL values from the second ART ANOVA analysis, which examines the contribution of individual temperature metrics and the interactions among them to the prediction residuals by the general temperature-$\Phi_{PSIImax}$ function derived using all data in the field site sub-dataset (**Lines 264-266, tracked version lines: 279-281**). Therefore, a clear legend of this second ART ANOVA in Figure 5 (Fig. D1 now) should be "ANOVA$_{RS\_gen}$", which refers to ART ANOVA using prediction residuals from general (non-PFT specific) temperature-$\Phi_{PSIImax}$ function.

Comparing the estimated aL values from the two ANOVA analyses in Figure 5 (now Fig. D1) aims to examine whether the contributions of individual temperature metrics and their interactions with prediction errors are consistent and to provide justification and flexibility for applying either version of aL values for CTI estimations. Now, the discussion of Figure 5 has been moved to **Appendix D**.

As the two versions of ANOVA analysis showed consistency, we applied aL values from the PFT-specific prediction residuals-based ART ANOVA (green bars in Fig. D1) to estimate CTI values corresponding to each $\Phi_{PSIImax}$ value in the field site sub-dataset, using Eq. 7. Then, these CTI values are incorporated to quantify the dependence of the parameters from PFT-specific temperature-$\Phi_{PSIImax}$ functions ($m_1$, $m_2$, $s_1$, $s_2$) on CTI, using quantile system approach (QSA)

(Section 2.3.4). For each parameter ($m_1$, $m_2$, $s_1$, $s_2$), we generated one CTI-informed function using all data from the field site sub-dataset. The reason for generating one equation for each parameter using all data from the field site sub-dataset is as follows. (1) A lack of sufficient data covering a large range of CTI values for any one PFT in the field site sub-dataset meant that CTI-dependence of each parameter ($m_1$, $m_2$, $s_1$, $s_2$) for each PFT could not be determined in a statistically robust way. (2) Moreover, the use of a single equation and comparison to the PFT-specific temperature-$\Phi_{PSIImax}$ functions can test the core hypothesis in this study: "climatological temperature regulates the temperature tolerance and resilience of $\Phi_{PSIImax}$, therefore shifts different PFT's temperature-$\Phi_{PSIImax}$ responses toward converged responses to the climatology of their "similar" local habitat."

In summary, we accepted the reviewer's comments and have clarified this concern by revising the following contents of the manuscript.

1) We have refined the description of the hypothesis and its testing methods.

**On lines 217-227 (Tracked version lines 225-235):** "To test the hypothesis that climatological temperature regulates the temperature tolerance and resilience of $\Phi_{PSIImax}$, and therefore shifts different PFT's temperature-$\Phi_{PSIImax}$ responses toward converged responses to the climatology of their "similar" local habitat, we generated a general climatology-informed temperature-$\Phi_{PSIImax}$ function and compared its results with the corresponding PFT-specific model results. In detail, we quantified corresponding climatological temperature metrics for data within the field site sub-dataset (Sect. 2.3.1) and assessed their capacity to explain the prediction residuals from PFT-specific temperature-$\Phi_{PSIImax}$ functions using ART ANOVA (Sect. 2.3.2). Based on the results, we incorporated the metrics via a linear combination into a Climatology Temperature Index (CTI) (Sect. 2.3.3). This index was then incorporated to quantify a CTI-informed temperature-$\Phi_{PSIImax}$ function (Sect. 2.3.4). The fitting results of this CTI-informed model were compared to the corresponding PFT-specific model results. Finally, we identified where prediction deficiency was improved by the CTI-informed parameterization and the climatology's effect on the temperature-$\Phi_{PSIImax}$ relationship was important to consider (Sect. 2.3.5)."

2) We have refined the description of the reasons for generating one equation for each parameter using all data from the field site sub-dataset after lines 285-286.

**On lines 297-303 (Tracked version lines 312-318):** "Ideally, the field site sub-dataset would cover diverse climatological temperature conditions, be distributed consistently across the full global range of CTI values, and contain statistically sufficient data for all PFTs, but this is not the case. The available 709 measurements represent a limited, non-uniform range of climatology temperature metrics (Histogram distribution of data in Fig. 2b). We overcome this data limitation by generating one CTI dependence function for each parameter in Eq. 1 using all data from the field site sub-dataset and the quantile system approach (QSA), which was developed to navigate the small sample size and inconsistent CTI values distribution by performing the following three steps."

3) To avoid confusion about the two versions of ART ANOVA and clarify the specific ART ANOVA finally employed to generate CTI-informed parameterization, we have revised their description in Figure D1 (original Figure 5) and the corresponding texts in Appendix D (Original section 3.2.1).

**"Appendix D | The contribution of climatology temperature metrics and derived CTI to prediction residues by the PFT-specific temperature-$\Phi_{PSIImax}$ functions**

There were consistent results between the ART ANOVA analysis for prediction residues estimated by the PFT-specific temperature-$\Phi_{PSIImax}$ functions (ANOVA$_{RS\_pft}$) and the general temperature-$\Phi_{PSIImax}$ function that resulted from fitting all data within the field site sub-dataset (ANOVA$_{RS\_gen}$) (Fig. D1). The ANOVA$_{RS\_gen}$ analysis showed that around 94% of variances in prediction residues by the PFT-specific temperature-$\Phi_{PSIImax}$ functions were able to be attributed between three climatological temperature metrics (WMET, SMET, and AAT) and their cross-terms. The interaction of WMET, SMET, and AAT showed the largest impact and explained around 37% of variations in prediction residues, followed by the cross effect of WMET and AAT with around 20% of variations in prediction residues associated with it (Fig. D1). In addition, the cross effect of SMET and WMET and the cross effect of SMET and AAT explained a similar amount (around 10%) of variations in prediction residues. Compared with the cross effect of three metrics, the main effects of individual metrics were relatively lower, with 8.78%, 6.42%, and 1.28% of variations in prediction residues associated with WMET, AAT, and SMET, respectively (Fig. D1). Similar to ANOVA$_{RS\_gen}$ analysis, the ANOVA$_{RS\_pft}$ analysis indicated that WMET, SMET, and AAT, as well as their interactions, explained 96% of variances in prediction residues by the PFT-specific temperature $\Phi_{PSIImax}$ function (Fig. D1).

[Figure]

Figure D1. Results comparison of two ART ANOVA analyses. ANOVA$_{RS\_pft}$ refers to results from ART ANOVA using prediction residuals from PFT-specific temperature-$\Phi_{PSIImax}$ functions. ANOVA$_{RS\_gen}$ refers to results from ART ANOVA using prediction residuals from the general (non-PFT specific) temperature-$\Phi_{PSIImax}$ function. Here WMET and SMET refer to the median

experienced temperature in the winter and summer respectively, while AAT refers to the annual average temperature."

3.    *Rearranging Section 3.2 and Section 3.3 and putting the CTI map (Figure 8c) before Figures 6 and 7 may help the audience more easily interpret results related to CTI.*

**Responses:** We agree with the reviewer that this rearranging of sections and figures can help interpret the main results related to CTI and benefit our readers by capturing key points of this study. To address this, we have rearranged sections 3.2 and 3.3 and the corresponding figures, with more details to be found in the response to **(5)** below.

4.    *Are the CTIs in the results section the general CTI?*

**Responses:** No, all CTI values after 3.2.1 were generated using the aL weights estimated from ART ANOVA using prediction residuals from the PFT-specific functions. As discussed in our responses to **Comment #2**, this is a point that we should have made clearer, and we have added specific language to address this in Appendix D as below.

**On Lines 864-865 (Tracked version lines 957-958):** "This consistency justified that the regulation of climatological temperature on the temperature-$\Phi_{PSIImax}$ relationship can be estimated using the results of either version of ANOVA. Here we will use the results from ANOVA$_{RS\_pft}$."

5.    *The manuscript is quite long, I suggest cutting the length of the manuscript. Some method and results could potentially be moved to the supplementary. For example, details of ART ANOVA, section 3.2.1, and section 3.2.3.*

**Responses:** This comment helps us polish our manuscript. Following this comment, a detailed discussion of ART ANOVA results, sections 3.2.1 and 3.2.3, including original Figure 5 and Figure 7, have been moved to Appendix D.  In the meantime, we have combined the original sections 3.2 and 3.3 and reorganized the text. In detail, section 3.2 is titled "Climatology's influence on the temperature-$\Phi_{PSIImax}$ relationship.", which aims to test the hypothesis that climatological temperature shifts different PFT's temperature-$\Phi_{PSIImax}$ responses toward converged responses to the climatology of their "similar" local habitat. Section 3.2 includes three sub-sections:

**3.2.1 CTI global pattern and its regulation on the temperature tolerance and resilience of $\Phi_{PSIImax}$ values.**

**3.2.2 Latitudinal variation in CTI-informed temperature tolerance and resilience of plant $\Phi_{PSIImax}$**

**3.2.3 Spatial distribution of the differences between CTI-informed and PFT-specific parameterizations**

Section 3.2.1 includes the description of the CTI global pattern (original section 3.3.1) and the result of CTI-informed temperature regulation on $\Phi_{PSIImax}$ (Original section 3.2.2). Then, this section briefly identifies the CTI range with improved prediction residues, compared with the prediction by the PFT-specific functions (A high-level summary of the original section 3.2.3). Instead, all details in two versions of ANOVA calculations and resultant CTI estimation and the original discussion of Figure 5 and Figure 7 are moved to Appendix D.

Section 3.2.2 describes the latitudinal variation of CTI-informed temperature tolerance and resilience of plant $\Phi_{PSIImax}$, the same as the original section 3.3.2.

Section 3.2.3 describes the spatial distribution of the differences between CTI-informed and PFT-specific parameterizations, the same as the original section 3.3.3).

**All revisions are on Lines 449-600 (Tracked version lines 469-689) and Appendix D 844-881 (Tracked version lines 936-974).**